

# Finiteness and the emergence of dualities

**Matilda Delgado[1,2]\*, Damian van de Heisteeg[2]†, Sanjay Raman[2]‡,
Ethan Torres[3]∘, Cumrun Vafa[2]§ and Kai Xu[2]¶**

**1** Max-Planck-Institut für Physik (Werner-Heisenberg-Institut),
Boltzmannstr. 8, 85748 Garching, Germany
**2** Jefferson Physical Laboratory, Harvard University,
17 Oxford St, Cambridge, MA 02138, USA
**3** Theoretical Physics Department, CERN,
1211 Geneva 23, Switzerland

⋆ matilda@mpp.mpg.de , † dvandeheisteeg@fas.harvard.edu , ‡ sanjayraman@fas.harvard.edu ,
∘ ethan.martin.torres@cern.ch , § vafa@g.harvard.edu , ¶ k_xu@g.harvard.edu

## Abstract

We argue that the finiteness of quantum gravity amplitudes in fully compactified theories (at least in supersymmetric cases) leads to a bottom-up prediction for the existence of non-trivial dualities. In particular, finiteness requires the moduli space of massless fields to be compactifiable, meaning that its volume must be finite or at least grow no faster than that of Euclidean space. Moreover, we relate the compactifiability of moduli spaces to the condition that the lattice of charged objects transform in a semisimple representation under the action of the duality group. These ideas are supported by a wide variety of string theory examples.

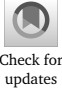

# 1  Introduction

One of the key facts about quantum gravity (QG) theories is the prevalence of non-trivial dualities [1]. Even though there is an abundance of evidence for these symmetries, there is still a lack of a clear bottom-up explanation for their existence. The aim of this work is to take a step towards exactly this.

Dualities manifest as we move around a moduli space of vacua, parametrized by the vacuum expectation values (vevs) of the massless scalar fields of the effective field theory (EFT). In particular, at asymptotic limits of the moduli space, the Distance Conjecture of [2] tells us that a tower of states emerges which becomes light exponentially with distance. Accordingly, it is also conjectured that the species scale [3,4], $\Lambda_s(\phi)$, decays to zero exponentially in the asymptotic regime of moduli space. (See [5–19] for studies of moduli space dependence of species scale.) In particular, for a fixed cutoff $\Lambda$ for the EFT, one must restrict the moduli space to a subset $\mathcal{M}_\Lambda$ for which $\Lambda_s(\phi) > \Lambda$. It was then argued in [20] that the truncated moduli space $\mathcal{M}_\Lambda$ at finite $\Lambda$ has finite volume:

$$V(\mathcal{M}_\Lambda) < \infty \, . \tag{1}$$

The argument in [20] proceeded by demanding finiteness of QG amplitudes as the theory is compactified to 1 dimension. Already, this indicates that the appearance of light towers at infinite distances is tied to the *principle of finiteness* in quantum gravity. In this paper, we introduce a condition which further refines Eq. (1), and we loosely term this condition 'compactifiability'. In particular, we propose that finiteness of amplitudes in QG requires that either the total moduli space volume *at* $\Lambda = 0$ be finite (in that $\lim_{\Lambda \to 0} V(\mathcal{M}_\Lambda) < \infty$) or at most that it grow no faster than the volume of Euclidean space with distance. That is, for $n = \dim(\mathcal{M}_\Lambda)$:

$$V(\mathcal{M}_\Lambda) \ll |\log \Lambda|^{n+\epsilon} \quad \text{as} \ \ \Lambda \to 0, \tag{2}$$

for arbitrarily small $\epsilon > 0$. Note also that Eq. (2) is closely related to the volume growth of $\mathcal{M}$ with geodesic distance. Indeed, assuming that $\Lambda$ goes exponentially to zero at large distances, the condition Eq. (2) reduces to the following:

$$V(\mathcal{M}_D) \ll D^{n+\epsilon} \quad \text{as} \quad D \to \infty, \tag{3}$$

where $\mathcal{M}_D$ are the points on the moduli space up to a distance $D$ away from an arbitrary fixed point. This is precisely the condition that the volume of $\mathcal{M}_D$ grow no faster with $D$ than that of a geodesic ball of radius $D$ in Euclidean space.

Our conjectures Eqs. (2) and (3) are motivated by a wealth of examples from string theory compactifications. For effective supergravity theories with 8 supercharges, we explain that the finiteness of volume of the vector multiplet moduli space is argued in geometric examples by the algebro-geometric notion of compactifiability as described in the mathematics literature. (Incidentally, this is in part the reason we call Eq. (2) the "compactifiability" condition.) We also argue from the bottom up that Eq. (2) follows from the finiteness of QG amplitudes (at least in supersymmetric theories): For a would-be non-compactifiable moduli space, we argue that there must necessarily arise an infinite number of massless normalizable ground states in the 1d supersymmetric quantum mechanics obtained upon compactification of all spatial dimensions. In geometric language, this is equivalently the statement that a would-be non-compactifiable moduli space would have infinitely many harmonic forms.

Furthermore, we also explain that the compactifiability of moduli space has implications for the action of duality group on the charged spectrum of the theory. In particular we argue that Eq. (2) implies that the representation of the duality group on physical degrees of freedom must be *semisimple*. For example, this precludes a situation where the $T$-transformation $T : \tau \to \tau + 1$ acting on the upper half-plane $\mathbb{H}$ is the only generator of a duality group. Contrast this with the case of Type IIB string theory in 10 dimensions, where the duality group is instead $SL(2, \mathbb{Z})$, generated by both the $T$- and $S$-transformations. We will discuss this example in detail throughout Sec. 2.

The organization of this paper is as follows. In Section 2, we set the stage: We first give a precise definition of duality groups and discuss their significance in string theory and in the Swampland Program. We then dissect the action of the duality group on the charged spectrum of the theory, and discuss how charged objects appear always to transform in semisimple representations of the duality group. In Section 3, we illustrate our discussion from Sec. 2 in a series of string-theoretic examples, identifying in each case how the duality group acts on the marked moduli space and the spectrum. Then, in Section 4, we present the notion of compactifiability. We further argue that compactifiability (or an algebraic definition thereof) of these spaces implies the semisimplicity of the representations under which the charged states transform in certain examples. We consider in particular compactifications of type II string theory on a Calabi-Yau threefold (CY3), where this can be proven explicitly. We also provide further evidence that compactifiability implies semisimplicity of representations by producing explicit putative quantum gravity moduli spaces that violate the semisimplicity condition and showing that compactifiability is also violated. Finally, we provide a bottom-up argument for the compactifiability of moduli spaces in Section 5. More specifically, we show that a violation of the compactifiability condition leads to an infinite number of degenerate vacua upon compactification to 1d, violating the *principle of finiteness* in quantum gravity. Finally, Section 6 contains our conclusions.

## 2 Moduli spaces and duality groups

In this section, we outline some basic facts about dualities and moduli spaces of vacua in quantum gravity. We start with giving a precise definition of a *duality group* and reviewing some general Swampland principles surrounding duality groups in quantum gravity. We will find it convenient to present our ideas through the important example of Type IIB string theory in 10 dimensions, which we first introduce in Sec. 2.1 and to which we will return several times over the course of this paper. In Sec. 2.2, we review several Swampland conjectures concerning the geometry and topology of moduli spaces in consistent EFTs coupled to gravity. These principles provide context and motivation for compactifiability, which will be discussed in Secs. 4 and 5.

We then detail the action of the duality group on the spectrum of charged $p$-brane objects in Sec. 2.3. We first discuss the "regular" case of electrically charged objects for $0 \le p \le d-4$, which form a discrete *charge lattice*. We then address the "special" cases of instantons ($p = -1$) and their magnetic duals, *duality vortices* ($p = d-3$). These objects constitute in some sense an "exceptional" type of duality actions. We introduce the notion of *semisimplicity* in Sec. 2.4, which appears to characterize the representations of duality groups on charge lattices. This will again be useful again in Secs. 4 and 5 when connecting the compactifiability of moduli space to the semisimplicity of representations of the duality group.

To begin, let us restrict our attention to consistent EFTs coupled to gravity with some sector of massless scalar fields (with no potential). In other words, there is an exact moduli space of vacua which is identical to the target space of the scalars modulo (duality) gauge transformations. (We will briefly comment on the effects of adding scalar potentials to the EFT in the conclusions.) We denote by $\mathcal{M}$ the moduli space parametrized by inequivalent vacuum expectation values (vevs) of the massless scalar fields in the action.

For a given quantum gravity vacuum, let $G^{(p)}$ denote its $p$-form gauge symmetry for $p \ge 0$, which we assume to be invertible (the invertibility of gauge symmetry in quantum gravity is well-motivated; see [21] for details and [22] for a related discussion of non-invertible worldsheet symmetries). For instance, any $p$-form RR gauge field in type II string theory is the potential for some $U(1)^{(p)} \subset G^{(p)}$, and the associated electrically- and magnetically-charged objects are the D-branes. Duality symmetries in QG are also gauge symmetries: two backgrounds related by a duality transformation are considered gauge-equivalent. Moreover, duality groups are 0-form gauge symmetries in the sense that one can consider non-trivial duality backgrounds as one winds around a 1-cycle. Indeed, the form-degree of the duality symmetry is fixed by the fact that its magnetically charged objects are codimension-2 vortices (we discuss this point further in Sec. 2.3.2). Additionally, since dualities generally act non-trivially on the vacua of the theory (as we will see in Sec. 2.2.1), they are generically *spontaneously-broken* 0-form gauge symmetries.

We now define the duality group, which we will henceforth denote $\Gamma$, to be the group of connected components of the 0-form (i.e. standard) gauge group.

$$\Gamma := \pi_0\left(G^{(0)}\right). \tag{4}$$

While the motivation of this definition will hopefully be made clear over the course of this section, one reason why we identify $\Gamma$ with $\pi_0\left(G^{(0)}\right)$ instead of the full $G^{(0)}$ is that $\Gamma$ is often considered to be a discrete group. For instance, in IIA string theory one has a RR 1-form potential $C_1$ and correspondingly a $U(1)^{(0)}$ gauge group factor, but such gauge transformations $C_1 \to C_1 + d\lambda$ are not typically what one would call a duality group action. On the other hand, continuous group actions that do appear as duality action, such as the the $SL(2, \mathbb{R})$ symmetry of 10D IIB supergravity, are always broken to discrete subgroups due to the presence of instantons.

As is well known, 0-form symmetries need not be abelian [23]. Thus, $\Gamma$ will generally take the form of a discrete, non-abelian group. Moreover, $\Gamma$ will in general act on the other $p$-form gauge symmetries of the theory via automorphisms for $p \geq 0$. In particular, $\Gamma$ acts on the various $p$-form gauge potentials and $G^{(p)}$-charged objects, which means these objects furnish representations of $\Gamma$ as we will discuss in greater detail in Sec. 2.3.1. For $p = 0$, the charged objects under $G^{(0)}$ need not organize themselves in a linear representation, but they still have an associative composition law endowing them with the structure of a group which is further acted on by $\Gamma$. We will discuss this point in further detail in Sec. 2.3.2. This $\Gamma$-action on the charged spectrum of the theory is thus represented by maps from $\Gamma$ to the automorphism groups of the various $p$-form groups:

$$\Gamma \to \mathrm{Aut}\left(G^{(p)}\right), \qquad p \geq 0.$$ (5)

Additionally, one can think of the action of $\Gamma$ on the moduli space scalars as the $p = -1$ analog of Eq. (5).

Throughout this section, we will find it useful to use the familiar example of the $\mathrm{SL}(2, \mathbb{Z})$ duality in Type IIB string theory in 10 dimensions as a working example. We now briefly review this construction.

## 2.1 Type IIB: A motivating example

Let us consider now a simple example of duality action which we will reference throughout the course of this paper. We will study Type IIB string theory in 10 dimensions, which is equipped with $S$-duality symmetry. The massless field content of Type IIB supergravity includes a complex scalar axiodilaton $\tau = C_0 + ie^{-\phi}$, NSNS and RR 2-form potentials $B_2, C_2$ and their magnetic duals, and a 4-form potential $C_4$ with self-dual field strength. The UV-complete theory is expected to enjoy a (spontaneously broken) $\mathrm{SL}(2, \mathbb{Z})$ 0-form gauge symmetry acting on the massless fields as follows:

$$\begin{bmatrix} a & b \\ c & d \end{bmatrix}: \ \tau \mapsto \frac{a\tau + b}{c\tau + d}, \qquad \begin{bmatrix} B_2 \\ C_2 \end{bmatrix} \mapsto \begin{bmatrix} a & b \\ c & d \end{bmatrix} \begin{bmatrix} B_2 \\ C_2 \end{bmatrix}, \qquad C_4 \mapsto C_4.$$ (6)

Thus, we say that the *duality group* $\Gamma$ of Type IIB is $\mathrm{SL}(2, \mathbb{Z})$.[1] The duality group is then seen to act on the *moduli space of vacua* $\mathcal{M}$ parametrized by the inequivalent values of $\tau$.

As seen in (6), the duality group acts linearly on the two-component vector of 2-forms and 6-forms, and thus also on the charges of $(p, q)$ strings and 5-branes. Each element of $\mathrm{SL}(2, \mathbb{Z})$ is mapped to a $2 \times 2$ matrix which then acts on $(B_2, C_2)^T$ (and $(B_6, C_6)^T$) as an automorphism of 2-forms and 6-forms. For example, the element of $\mathrm{SL}(2, \mathbb{Z})$ that implements the strong-weak coupling duality $g_s \mapsto 1/g_s$ exchanges the $B_2$ and the $C_2$ gauge fields. Notice that in general the map (5) need not be injective. For instance, all elements of $\mathrm{SL}(2, \mathbb{Z})$ leave the RR 4-form invariant. Similarly, the action on the moduli space $\mathcal{M}$ itself will generally factor through a quotient $\Gamma/H$ for some normal subgroup $H \subset \Gamma$ under which $\mathcal{M}$ is invariant. In this case, the center $Z = \mathbb{Z}_2 = \{\pm 1\} \subseteq \mathrm{SL}(2, \mathbb{Z})$ acts trivially on the moduli, so the action of $\Gamma$ on the moduli factors through $\mathrm{PSL}(2, \mathbb{Z}) = \mathrm{SL}(2, \mathbb{Z})/\mathbb{Z}_2$. Note however that this $\mathbb{Z}_2$ does not leave the

---

[1]The duality group $\mathrm{SL}(2, \mathbb{Z})$ is technically only a subgroup of the full duality group that acts on the massless spectrum. To obtain the full duality group, one also needs to take into account the duality action on fermions, which centrally extends the group by worldsheet fermion parity reversal $(-1)^F$. Moreover, the $\mathbb{Z}_2$ gauge symmetry action $\Omega$, which is perturbatively realized as F1 worldsheet orientation reversal, further extends the full duality group is $\mathrm{GL}^+(2, \mathbb{Z})$, which is defined to be the $\mathrm{Pin}^+$-cover of $\mathrm{GL}(2, \mathbb{Z})$ [24, 25]. Note that $\mathrm{GL}^+(2, \mathbb{Z})$ factors through $\mathrm{PGL}(2, \mathbb{Z})$ when acting on the axiodilaton, with the $\mathbb{Z}_2$ factor acting via the $\tau \mapsto -\overline{\tau}$. This technicality is tangential to our main points so we will only discuss the $\mathrm{GL}^+$-lift of the duality group sparingly in the remainder of this work. Indeed, the additional elements in the $\mathrm{GL}^+$-lift of the duality group do not act on moduli space and therefore do not affect the compactifiability criterion.

charges of $(p,q)$-strings and 5-branes invariant, so the full $SL(2,\mathbb{Z})$ is indeed seen to act on the spectrum of charged states.

An important feature of this discussion is that at a generic vacuum value $p \in \mathcal{M}$, $\Gamma$ will in general be spontaneously broken to a subgroup (called by definition the *stabilizer group* $\mathrm{Stab}_p(\Gamma)$ associated to $p$). In Type IIB, $\Gamma = SL(2,\mathbb{Z})$ is spontaneously broken to its central $\mathbb{Z}_2$ everywhere in $\mathcal{M}$ away from the *duality fixed points* $\tau = i$ and $\tau = e^{2\pi i/3}$, at which it is broken to larger finite groups. Specifically, the generators fixing $\tau = i$ and $\tau = e^{2\pi i/3}$ are $S$ and $U = ST$ respectively which are

$$S = \begin{bmatrix} 0 & -1 \\ 1 & 0 \end{bmatrix}, \qquad U = \begin{bmatrix} 0 & -1 \\ 1 & 1 \end{bmatrix}, \tag{7}$$

in the standard 2-dimensional matrix representation. At $\tau = i$ and $\tau = e^{2\pi i/3}$, $SL(2,\mathbb{Z})$ is therefore broken to the cyclic subgroups $\mathbb{Z}_4, \mathbb{Z}_6 \subset SL(2,\mathbb{Z})$ generated by $S, U$, respectively.

Equipped with this illustrative example, we now turn to reviewing some existing Swampland conjectures concerning properties of moduli spaces and duality groups.

## 2.2 The geometry of moduli spaces

In this section, we will briefly review a series of Swampland conjectures about the geometry of moduli spaces. These conjectures provide context and motivation for our subsequent study of the compactifiability of moduli spaces.

### 2.2.1 Marked moduli space conjecture

In this section, we briefly review the results of [26] governing the global geometry of moduli spaces of vacua and the action of duality groups on them. Fix an EFT coupled to gravity (admitting a consistent UV-completion) which has a moduli space of vacua $\mathcal{M}$. Motivated by completeness of the spectrum and a wide assortment of supersymmetric examples, the authors of [26] introduced the notion of a *marked moduli space* $\hat{\mathcal{M}}$ which makes physical the notion of a covering space over $\mathcal{M}$. More precisely, $\hat{\mathcal{M}}$ is defined to be the space whose points specify vacua of the theory along with a basis for its observables.

For example, let us take Type IIB string theory in 10 dimensions.[2] Its massless scalar field content consists precisely of the axiodilaton $\tau$, which *a priori* takes values in the upper half plane $\mathbb{H}$ of complex numbers with positive imaginary part. Its moduli space of vacua $\mathcal{M}$ is then given by the quotient $SL(2,\mathbb{Z})\backslash\mathbb{H}$ of the upper half-plane by the action of the $S$-duality group $SL(2,\mathbb{Z})$. However, the theory is also equipped with a spectrum of $(p,q)$-strings and 5-branes on which $S$-duality acts faithfully in the standard 2-dimensional matrix representation, as discussed. The lattices spanned by the $(p,q)$-strings and 5-branes (which will be discussed in greater detail in 2.3.1) can therefore be thought to be fibered over each point in $\mathcal{M}$. Choosing a basis for the charge lattice therefore amounts to passing to the covering space of $\mathcal{M}$ (on which the duality group acts by deck transformation). This is nothing but the upper half-plane $\mathbb{H}$ itself. In this example, the marked moduli space $\hat{\mathcal{M}}$ is the naïve space parametrized by the scalars of the theory before one takes a quotient by the duality group action—in Type IIB, this is precisely the space $\mathbb{H}$.

Given the definition of the marked moduli space, it was then argued in [26] that the following should hold:

**Conjecture 2.1** (Marked Moduli Space Conjecture)**.** *Let $\hat{\mathcal{M}}$ be the marked moduli space of a consistent EFT coupled to gravity. Then $\hat{\mathcal{M}}$ is contractible.*

---

[2]See [27] for a recent test of the marked moduli space conjecture in 5d M-theory and 4d $\mathcal{N} = 2$ CY3 compactifications.

This is clearly satisfied by $\hat{\mathcal{M}} = \mathbb{H}$ as discussed previously and is observed to hold in all known string theory examples with a moduli space. Note also that Conjecture 2.1 can be rephrased as the statement that $\mathcal{M}$ is necessarily realizable as the quotient $\Gamma'\backslash\hat{\mathcal{M}}$ of a contractible space by a group $\Gamma'$, where $\Gamma'$ (which is possibly a quotient of $\Gamma$) acts non-trivially on $\mathcal{M}$. This means that the part of the duality group that acts non-trivially on moduli space can in some sense be *identified* with the topology of $\mathcal{M}$:[3]

$$\pi_1^{\mathrm{orb}}(\mathcal{M}) \simeq \Gamma'. \tag{8}$$

Here, $\pi_1^{\mathrm{orb}}$ is the *orbifold fundamental group*[4] which takes into account the fixed points of a group action on $\hat{\mathcal{M}}$.

The notion of an orbifold fundamental group is standard in the mathematics literature but not in the physics literature, so let us take a moment to explain the intuition behind its construction. In particular, let $\mathcal{M}$ be an orbifold, which can be thought of as a topological space looking locally almost everywhere like $\mathbb{R}^n$, except at isolated orbifold singularities, labeled by an index $i$. At these orbifold loci, the space looks locally like $\mathbb{R}^{n-m_i} \times \mathbb{R}^{m_i}/G_i$ for $G_i$ a discrete group which acts discontinuously. If all of these singularities have codimension greater than two, i.e. all $m_i > 2$, then the orbifold fundamental group is given as follows:[5]

1. Delete the orbifold singularities of $\mathcal{M}$ to obtain a space $\mathcal{M}^\circ$.

2. Then $\pi_1^{\mathrm{orb}}(\mathcal{M}) := \pi_1(\mathcal{M}^\circ)$.

Note that this formula is a refinement to orbifolds of a theorem due to Armstrong [30]: If a discrete group $G$ acts discontinuously on a simply-connected metric space $X$, and $H \subset G$ is the normal subgroup generated by elements with fixed points, then $\pi_1(X/G) \simeq G/H$. For the case of Type IIB in 10 dimensions, there are two nontrivial orbifold points which locally look like $\mathbb{R}^2/\mathbb{Z}_3$ and $\mathbb{R}^2/\mathbb{Z}_2$, respectively. Since these orbifold singularities are of codimension two, we cannot simply delete the associated loci and compute the fundamental group of the resulting $\mathcal{M}^\circ$. Instead, we associate to each orbifold fixed locus that locally looks like $\mathbb{R}^2/G_i$ a factor of $G$. The associated orbifold fundamental group is then given as a free product over the $G_i$:

$$\pi_1^{\mathrm{orb}}(\mathcal{M}) = \mathbb{Z}_2 * \mathbb{Z}_3 = \mathrm{PSL}(2,\mathbb{Z}), \tag{9}$$

which is precisely the group which acts on the marked moduli space $\hat{\mathcal{M}}$, the quotient by whose action is the moduli space $\mathcal{M}$.

Comparing Eq. (8) with Eq. (4), we seem to have two complementary definitions of the duality group.[6] To illustrate relation between the two, consider a theory with a single periodic scalar which can be thought of as a gauge potential for a $(-1)$-form gauge symmetry. This theory also possesses a $\mathbb{Z}^{(0)}$ 0-form gauge symmetry corresponding to integer shifts of the scalar. In particular the relation

$$\pi_1\left(U(1)^{(-1)}\right) = \pi_0\left(\mathbb{Z}^{(0)}\right) = \mathbb{Z}, \tag{10}$$

ties together both ways of seeing that the duality group in this case is $\mathbb{Z}$. Note that more generally, we can identify the continuous $(-1)$-form gauge symmetry of the theory with the moduli space $\mathcal{M} = \mathcal{M}^{(-1)}$, which of course need not even be a group.

---

[3]Although taking a quotient of $\hat{\mathcal{M}}$ by the action $\Gamma'$ can in general lead to non-trivial $\pi_n(\mathcal{M})$ with $n > 1$, we argue that $\Gamma'$ itself can be obtained from $\pi_1^{\mathrm{orb}}(\mathcal{M})$.

[4]In the case that $X$ is an algebraic stack (which will cover essentially all examples of interest), the orbifold fundamental group $\pi_1^{\mathrm{orb}}(X)$ can be identified with the (pro-)étale fundamental group $\pi_1^{\mathrm{ét}}(X)$.

[5]For instance, see page 12 of [28], as well as [29] for a helpful review of orbifold homotopy groups. When $\mathcal{M}$ has codimension-2 fixed points, then one cannot simply declare $\pi_1^{\mathrm{orb}}(\mathcal{M}) := \pi_1(\mathcal{M}^\circ)$ because we lose information due to the fact that $\pi_1(S^1/G) = \mathbb{Z}$ for any finitely freely acting $G$, whereas $\pi_1(S^{2k-1}/G) = G$ for $k > 1$.

[6]Albeit the former is only sensitive to the quotient group $\Gamma'$.

A key conceptual point to draw from this discussion is that the marked moduli space conjecture implies that topology of the moduli space $\mathcal{M}$ is inextricably tied to the duality group $\Gamma$ itself; this is made explicit through Eq. (8). In fact, the marked moduli space conjecture was initially motivated by topological triviality along the same lines as the cobordism conjecture, but it seems to be a somewhat orthogonal principle since it deals only with the geometry of the *vacua* of the theory.

It is important to note that the physics of Conjecture 2.1 is encapsulated in the definition of the marked moduli space. Importantly, the marked moduli space is not *defined* to be the universal covering space. Indeed, the identification of the marked moduli space with the covering space rests on the existence of objects in the theory transforming *faithfully* under the duality action. For this reason, the marked moduli space conjecture is motivated by completeness of the spectrum. Thus, the marked moduli space conjecture further motivates the study of the *action* of the duality group on charged objects in the theory: Is there additional information about the geometry of moduli space that is encoded within the duality group action (and vice versa)? The entirety of Sec. 2.3 is devoted to further exploring this question. For the remainder of this section, we will review further Swampland principles and string theory examples about the geometry of moduli space that set the stage for our later discussion in Secs. 2.3 and 4.

### 2.2.2 No-minimum-length conjecture and fixed points

In this section, we will review an older but related set of conjectures about the geometry of moduli spaces and duality actions, first developed in [2]. This work was the first to propose the well-known distance conjecture, which has been the basis for a remarkable explosion of recent developments in the Swampland program. Our main concern will be a different conjecture proposed in [2], the so-called "no-minimum-length"-conjecture:

**Conjecture 2.2** (No-Minimum-Length Conjecture). *Let $\mathcal{M}$ be the moduli space of a consistent EFT coupled to gravity. Then in each homology class in $H_1(\mathcal{M})$, there is no 1-cycle of minimum length.*

One has to be careful in stating the conjecture, as it should refer to *homology* cycles and not *homotopy* classes: there exist nontrivial homotopy classes which have curves of minimum length. (Somewhat confusingly, this conjecture is often referred to as the "$\pi_1$-conjecture" in the literature, though it should perhaps be called the "$H_1$-conjecture".) However, in all known examples, the offending homotopy classes map to zero in the abelianization $\pi_1(\mathcal{M}) \rightarrow H_1(\mathcal{M})$. We will discuss this point at the end of this section.

Here, the notion of *length* on $\mathcal{M}$ is specified by the physical metric derived from the scalar kinetic terms in the EFT action. Note also that the $H_1$ used in this conjecture is the *topological* $H_1$ of a topological space, which is distinct from the orbifold homotopy groups defined considered in the previous section. This distinction, as we will see, makes all the difference.

At first sight, it appears that this conjecture is closely related to the marked moduli space conjecture, since they both relate to triviality of topology in some sense. However, the no-minimum-length conjecture actually points to a very different (and complementary) feature of moduli space geometry. It argues that any loop in the *unmarked* moduli space must be "shrinkable" to arbitrarily small size,[7] independent of what the covering space looks like. Thus, the no-minimum-length conjecture gives a hint towards the action of the duality group on the marked moduli space in the following sense: It tells us that the duality group $\Gamma$ must act on $\hat{\mathcal{M}}$ in such a way that the topological space $\mathcal{M}$ underlying $\mathcal{M}$ has no nontrivial 1-cycles, at least

---

[7]The scare quotes refer to the fact that we allow for shrinking maneuvers in which closed loop may split apart into multiple closed loops along the way in a way that preserves its homology class. Equivalently, the loop is null-bordant inside $\mathcal{M}$.

when it is suitably compactified. This motivates consideration of the notion of *compactifiability* of moduli spaces, and compactifiability will indeed become relevant in our discussion in Secs. 4 and 5.

Let us consider a simple (non-)example. Consider the action of $\mathbb{Z}$ on $\mathbb{R}$ by the integer shift $x \mapsto x + 1$. Then $\mathbb{R}/\mathbb{Z} \simeq S^1$, for which $H_1(S^1) = \pi_1(S^1) \simeq \mathbb{Z}$, and in each homology (and homotopy) class of $\pi_1(S^1)$ there is clearly a curve of minimum length – (an appropriate multiple of) the $S^1$ itself. Thus, the moduli space $\mathcal{M} \simeq S^1$ is ruled out by the no-minimum-length conjecture, even if the corresponding marked moduli space is the universal cover $\hat{\mathcal{M}} = \mathbb{R}$ (which is *a priori* allowed). However, if one is able to "fill in" the circle, then the no-minimum-length conjecture is satisfied. This is, for instance, the case in Type IIB where taking the weak coupling limit causes the circle to shrink to zero size. Indeed, recall that the Type IIB axiodilaton is given by $\tau = x + ie^{-\phi}$, where we have let $x = C_0$ be the RR axion. The metric on the upper-half plane in these coordinates given by:

$$ds^2 = d\phi^2 + e^{-2\phi}dx^2 \,. \tag{11}$$

Note that the circle parametrized by the RR axion $x$ shrinks in the weak-coupling limit. Note also that the metric is the same on $\mathcal{M}$ and $\hat{\mathcal{M}}$, and the difference between the two is the global data given by the boundary conditions. Indeed, for Type IIB, the global geometry of $\mathcal{M}$ is encapsulated in (11) combined with the fact that $x$ is identified with $x + 1$ and that there is a boundary at $|\tau| = |C_0 + ie^{-\phi}| = 1$.

Equivalent to the no-minimum-length conjecture is the proposal that the duality group action on $\hat{\mathcal{M}}$ be generated by elements that have fixed points. To see the equivalence, note first that the marked moduli space conjecture implies, by Armstrong's theorem [30], that a free group action on $\hat{\mathcal{M}}$ would result in a nontrivial topological $\pi_1(\mathcal{M})$, the abelianization of which is $H_1(\mathcal{M})$. This contradicts the no-minimum-length conjecture (at least in the case that $\pi_1(\mathcal{M})$ has a nontrivial abelianization), so the group must therefore act with fixed points. Conversely, if the duality group acts with fixed points, then the no-minimum-length conjecture easily follows. Indeed, as noted above, $\Gamma$ can be understood as a spontaneously broken (0-form) gauge symmetry with various subgroups of $\Gamma$ restored at its fixed points in $\mathcal{M}$. In this case, by the marked moduli space conjecture, any loop in $\mathcal{M}$ can be realized as corresponding to an action by $g \in \Gamma$, which can then be decomposed into a product of generators $h_i$ which have fixed points. It would then be possible to "unhook" the loop around each fixed point $p_i$ associated to the $h_i$ and therefore contract the loop, as claimed by the no-minimum-length conjecture.

For finite cyclic groups acting on a finite-dimensional contractible space $\hat{\mathcal{M}}$, the Lefschetz fixed point theorem [31] tells us that such actions necessarily have fixed points in the interior of $\hat{\mathcal{M}}$. This therefore argues for the no-minimum-length conjecture in the case that the duality group $\Gamma$ is generated by elements of finite order. For the case of Type IIB, first recall that the group $SL(2, \mathbb{Z})$ factors through $PSL(2, \mathbb{Z}) = \mathbb{Z}_2 * \mathbb{Z}_3$ in its action on the marked moduli space $\hat{\mathcal{M}} = \mathbb{H}$. The discussion around Eq. (7) already shows then that the $\mathbb{Z}_2$ and $\mathbb{Z}_3$ generators of $PSL(2, \mathbb{Z})$ are respectively preserved at $\tau = i$ and $\tau = e^{2\pi i/3}$ respectively.

From the Type IIB example, and in a wealth of additional examples in the string landscape with at least 16 supercharges, one might conjecture that the duality group is always generated by elements of finite order. This is not at all true! In section 3.2, we will provide detailed examples of Type IIB compactifications on a Calabi-Yau threefold (CY3) for which the duality group is generated by elements of infinite order.

It is also in these examples that a formulation of the no-minimum-length conjecture in terms of the *homotopy* group $\pi_1(\mathcal{M})$ fails. Although we defer a detailed discussion to Sec. 3.2, we briefly explain this subtlety here. Consider Type IIB on the mirror quintic threefold; the exact vector-multiplet moduli space $\mathcal{M}$ is then the complex structure moduli space of the

mirror quintic. This moduli space has a finite-distance conifold singularity and an infinite-distance large-complex-structure (LCS) point, the monodromy group around both of which is $\mathbb{Z}$ acting on the charged states of the theory. Associated to each of these points there is a free factor $\mathbb{Z}$ in $\pi_1(\mathcal{M})$. Consider now a "figure-eight" path winding around both of these monodomy fixed points. Such a path is homologically equivalent to two loops winding around the two points, both of which are "shrinkable" to zero size. However, the figure-eight path is homotopically inequivalent to the two loops, and indeed cannot be shrunk to zero size within its homotopy class alone. Despite this subtlety, it still appears that the *homological* version of the no-minimum-length conjecture is satisfied by these (and all other) examples in string theory. Before describing these other examples in detail, however, we will make a precise description of the action of the duality group on the spectrum of charged objects.

## 2.3 Duality actions

In this section, we will study the action of duality groups on the charged spectrum of the theory. In the context of the marked moduli space conjecture, it is expected that a consistent EFT coupled to gravity has a charged spectrum which transforms faithfully under the action of the duality group. In practice, the duality group action manifests very differently in different examples, and the main purpose of this section will be to develop some intuition for the general form of duality group actions.

When studying the action of the duality group on charged objects, it makes sense to understand the following spacetime picture. For each element of a duality group $\Gamma$, there exists a codimension-2 dynamical vortex object in spacetime, which we will call a *duality vortex*. The duality vortex associated to $g \in \Gamma$ acts on a charged object $q$ in the theory via $q \mapsto gq$ as the object winds around the vortex. For example, in four dimensional theories, the duality vortex is just an axionic string in the special case where the duality group action is that of an integer shift of a periodic scalar field.

The existence of duality vortices in the EFT follows from the requirement that there be no exact global symmetries for the corresponding UV complete quantum gravity theory. In particular, it was argued in [32] (see also [33]) that for any UV complete quantum gravity vacuum with 0-form gauge group $G^{(0)}$, there must exist dynamical vortices with monodromies for each element $g \in \pi_0(G^{(0)})$. Since $\Gamma := \pi_0(G^{(0)})$ is our duality group by definition, we thus expect such duality vortices with the only difference from the considerations of [32] being that $\Gamma$ may be (partially) spontaneously broken.[8] All this means is that (a subset of the) duality vortices will induce a monodromy for the moduli space scalars around them.

To sketch the reasoning of [32], first recall that for a given gauge group $G^{(0)}$ there exists Wilson lines labeled by all representations of $G^{(0)}$. When $G^{(0)}$ has disconnected components, Wilson lines labeled by some $R \in \text{Rep}(\pi_0(G^{(0)}))$ will be topological. For instance if $G^{(0)}$ is discrete, the gauge degrees of freedom are purely topological, so clearly the Wilson lines will also be topological. Such topological lines generate a non-invertible $\text{Rep}(\Gamma)^{(d-2)}$ symmetry. The charged objects under these symmetries are Gukov-Witten (GW) operators which are codimension-2 defects (i.e. infinite tension objects) defined formally by declaring that there is a monodromy $g \in \Gamma$ around its normal directions [35,36]. The key point then is as follows: If the duality vortices are *dynamical*, then the GW operators can end on their (codimension-3) creation operators. Due to the link-pairing between the GW operators and Wilson lines, this implies that the $\Gamma$ Wilson lines can no longer be topological, breaking the would-be global symmetry. Equivalently, such duality vortices are seen to be required by magnetic (vortex) completeness for the $\Gamma$ gauge symmetry.

---

[8]For a spontaneously broken discrete gauge group is that one would expect that there are domain walls associated with each $g \in \Gamma$. However, the duality vortices will cause these domain walls to be unstable; see [34] for a recent discussion of this point.

The action of the duality group of a $d$-dimensional theory on the spectrum of charged objects can be split in two conceptually different cases. In the first case, the objects are charged under $(p+1)$-form gauge potentials (i.e. $p$-form gauge symmetry) with $0 \le p \le d-4$ on which the duality group acts. We will discuss this case first, in Sec. 2.3.1, where we will study the action of the duality group on objects of codimension $k = d - p - 1 > 2$. (When using the term "codimension-$k$", we will typically assume $k < d$.) In these examples, the duality group, which we will denote as usual by $\Gamma$, acts on an abelian category of representations of the gauge group of the theory. These representations will in general organize themselves in a *charge lattice* $\Lambda$. We therefore see that the representations of the duality group on codimension $k > 2$ objects are modules for the group algebra of the duality group over $\mathbb{Z}$; that is, they constitute *integer representations* of the duality group. We will observe in several examples that not just any integer representation of the duality group will suffice. In particular, the duality group must in some sense act "democratically" on elements of the charge lattice – it cannot mix elements from one sublattice into another without being able to go in the other direction.

The second case corresponds to the objects that are in a sense magnetically or electrically charged under the $(-1)$-form symmetry whose gauge potentials are the moduli fields themselves. That is, we consider duality actions on $p$-branes with $p = -1$ and $p = d-3$. These cases respectively correspond to codimension-2 objects (duality vortices) and their electric-magnetic duals, instantons. Indeed, in contrast with the previous case, the action of $\Gamma$ on codimension-2 charged objects in theory will generally not take the form of an integer representation. Indeed, the nontrivial monodromy of such charged objects around each other ensures that the fusion of codimension-2 defects need not be abelian. An example of duality action on codimension-2 charged objects is the action of duality vortices on themselves. In this case, the vortices generate a (generally nonabelian) group $\Gamma$ (and not an abelian charge lattice), and $\Gamma$ acts on itself in the adjoint. This is illustrated in the Figure 1. We will describe these duality vortices as well as their dual instantons in Sec. 2.3.2. For further motivating examples, see Sec. 4.3.

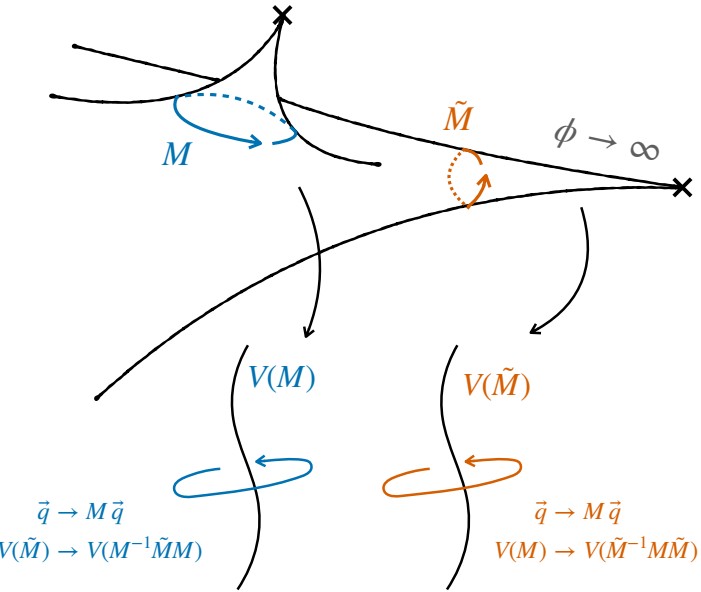

Figure 1: Two vortices are shown, associated to two monodromy actions of the duality group (one on the boundary of moduli space, one in the bulk), together with their action on other vortices and on the charges of objects of codimension greater than two.

Before moving on to a more detailed discussion of these two cases, let us briefly comment on the possibility of domain walls: these codimension-1 objects are charged under $(d-1)$-form gauge fields which may transform under the action of the duality group. For the most part, we will leave them out of our discussion because their presence generically generates a scalar potential for the moduli, breaking the exact $(-1)$-form gauge symmetry. Indeed, by definition, they source non-dynamical $(d-1)$-form gauge potentials whose kinetic terms lead to scalar potentials in the effective action. The simplest example of this is that of a D8 brane in Type IIA string theory which generates a run-away potential for the dilaton, parametrized by the Romans' mass [37]. Interestingly, requiring these potentials to be invariant under the action of the duality group can lead to a non-trivial mixing of the action of the duality group with that of the $(d-2)$-form symmetry, as we will see in an example in Section 3.2. We defer a detailed discussion of domain walls and how they transform under duality group to future work, as we will mostly focus on *exact* moduli spaces in what follows.

### 2.3.1  Duality action on codimension $2 < k < d$ objects

In this section, we describe the action of the duality group on codimension $2 < k < d$ objects. Let us denote $S_k$ as the collection of defects up to deformations that preserve its gauge charges. $S_k$ is equipped with an abelian fusion rule (we provide justification for this fact in section 2.3.2) which is respected by the action of the duality group. Thus, the set $S_k$ enjoys the structure of a commutative monoid equipped with an action by $\Gamma$. Moreover, there is strong evidence that in the context of quantum gravity, any non-invertible categorical fusion rule is broken to its maximal invertible subcategory by the inclusion of backgrounds of nontrivial topology [21,38]. As such, we expect $S_k$ to in fact be an abelian group.

Before we describe the general picture, let us once again consider our illustrative example of Type IIB in 10d. Recall that $SL(2,\mathbb{Z})$ duality acts on the $(p,q)$-strings and $(p,q)$-5-branes which couple electrically and magnetically to $pB_2 + qC_2$, where $B_2, C_2$ are the NSNS and RR 2-form gauge fields. The charge lattice in this case is isomorphic to $\mathbb{Z}^2$, and the duality acts via

$$\begin{bmatrix} p \\ q \end{bmatrix} \mapsto \begin{bmatrix} a & b \\ c & d \end{bmatrix} \begin{bmatrix} p \\ q \end{bmatrix}, \qquad \begin{bmatrix} a & b \\ c & d \end{bmatrix} \in SL(2,\mathbb{Z}), \tag{12}$$

on a $(p,q)$-string or $(p,q)$-5-branes. In this case, the $(p,q)$-strings and 5-branes (which are clearly of codimension at least 2 in the 10d effective theory) parametrize two copies of the charge lattice $\mathbb{Z}^2$. The duality group action therefore amounts to the natural action of $SL(2,\mathbb{Z})$ on $\mathbb{Z}^2$ in its 2-dimensional matrix representation. Note that although the elements of $SL(2,\mathbb{Z})$ that act non-trivially on $\tau$ are those of $PSL(2,\mathbb{Z})$, the spectrum of $(p,q)$-charged objects is acted on by the full $SL(2,\mathbb{Z})$.

More generally, consider again the set $S_k$ of defects in fixed codimension $k > 2$ charged under the duality action. As we have seen, $S_k$ enjoys the structure of an abelian group. Henceforth, we will therefore assume that the fusion rule of codimension $k > 2$ defects is captured by a *finitely-generated* abelian group, which is well-motivated by finiteness principles in quantum gravity. By the structure theorem for finitely-generated abelian groups, we therefore have that

$$S_k \simeq \mathbb{Z}^r \oplus \bigoplus_{k \geq 1} \mathbb{Z}_k^{n_k}, \tag{13}$$

where all but finitely many of the $n_k$ are zero. We see that $S_k$ is effectively the direct sum of a lattice and a finite abelian group parametrizing torsional charges. It would be interesting to explore possible constraints on the allowed values of $r$, $k$ and $n_k$ but we leave this for future work; nevertheless, for a step in that direction see Conjecture 2 in [39].

The example of torsion charges is interesting and shows up often in string theory examples. A simple instance of this occurs when the internal geometry has torsion cycles. Branes wrapping these cycles then lead to objects with torsion charges. Another instance involves D-brane charges in the presence of NS5 brane fluxes [40, 41]. In particular, if part of the string worldsheet is a WZW model with target $SU(2) \simeq S^3$ with $\int_{S^3} H_3 = N$, then the D-branes are classified by twisted K-theory, where the twist data is the NSNS flux. One can show that such D-branes are torsion charged, which agrees with the twisted K-theory groups $K_H^0(S^3) = K_H^1(S^3) = \mathbb{Z}_N$.

Let us therefore consider an action of a duality group $\Gamma$ on a free, finitely generated abelian group $\Lambda_k$ generated by codimension-$k$ defects. Then $\Lambda_k \simeq \mathbb{Z}^r$, so the objects transforming under duality organize in a *charge lattice* of rank $r$. Furthermore, the duality action constitutes a map $\Gamma : \mathbb{Z}^r \to \mathbb{Z}^r$, which therefore amounts to a map $\Gamma \to \text{Aut}(\mathbb{Z}^r)$ from $\Gamma$ to the (outer) automorphism group of $\mathbb{Z}^r$. This describes an *integer representation* of the duality group $\Gamma$, which can alternatively be understood as a free module over the group algebra $\mathbb{Z}[\Gamma]$.

With this framework in place, a natural next question is to understand possible Swampland constraints on the duality action on charge lattices $\Lambda_k$, imposed by quantum gravity and realized in string theory examples. For this purpose, integer representations remain a rather unwieldy tool, since their mathematical theory is relatively poorly developed by comparison to representation theory over $\mathbb{R}$ or $\mathbb{C}$. As such, it is useful to instead consider the *complex* representation theory of $\Gamma$ over $\Lambda_k \otimes \mathbb{C} \simeq \mathbb{C}^r$. The relevant question is now posed as follows: What Swampland constraints can be imposed on the complex representations of $\Gamma$ over $\Lambda_k \otimes \mathbb{C}$? This question will be answered in section 2.4. Before then, we turn to describing the action of the duality group on duality vortices and their dual instantons.

### 2.3.2 Duality action on vortices and instantons

We now turn to a general discussion of duality group actions on vortices and their dual instantons. Suppose we are given a theory with an exact moduli space $\mathcal{M}$ and a duality group $\Gamma$ which acts on the spectrum of the theory. In this section, we will develop a natural spacetime picture for the duality action.

Let's first consider duality vortices in our favorite example of Type IIB string theory in 10 dimensions. The duality vortices that correspond to each element of $SL(2, \mathbb{Z})$ can be understood in terms of a profile for the scalar fields of the theory, on which $S$-duality also acts. Namely, the low-energy effective field theory contains an exact complex modulus, the axiodilaton $\tau = C_0 + ie^{-\phi}$, which is a linear combination of the periodic RR 0-form gauge field $C_0$ and the dilaton $\phi$. The action of $SL(2, \mathbb{Z})$ on $\tau$ is via

$$\tau \mapsto \frac{a\tau + b}{c\tau + d}. \tag{14}$$

The moduli space, as discussed in Sec. 2.2, is the quotient of the upper half plane $\mathbb{H} = \hat{\mathcal{M}}$ of values parametrized by $\tau$ modulo $SL(2, \mathbb{Z})$:

$$\mathcal{M} = SL(2, \mathbb{Z}) \backslash \hat{\mathcal{M}}. \tag{15}$$

The effective field theory therefore specifies a map $X \to \mathcal{M}$, where $X$ is the *physical* spacetime configuration. For simplicity, let $X = \mathbb{R}^{9,1}$, and suppose we excise a codimension-2 locus $\Sigma \simeq \mathbb{R}^{7,1}$. Then $X \backslash \Sigma$ is homotopy equivalent to $S^1$, so at the level of topology, the effective field theory specifies a map $S^1 \to \mathcal{M}$, which corresponds to an element of the fundamental group $\pi_1(\mathcal{M})$. In fact, as we have seen, the correct object to consider is in fact the *orbifold* fundamental group $\pi_1^{\text{orb}}(\mathcal{M})$, since in general the duality action on the marked moduli space will have fixed points as discussed in 2.2.2.

For Type IIB, the covering $\hat{\mathcal{M}}$ is contractible. (The marked moduli space conjecture asserts that this is a general Swampland principle.) Therefore, a nontrivial map

$S^1 \to \mathcal{M} = \mathrm{SL}(2,\mathbb{Z})\backslash\hat{\mathcal{M}}$ then corresponds to a monodromy around the $S^1$ by an element of the duality group by which we quotient $\hat{\mathcal{M}}$. We conclude that there exists a solution of the effective theory corresponding to a codimension-2 locus about which there is a monodromy by an element of $\mathrm{SL}(2,\mathbb{Z})$. In particular, there exists a codimension-2 duality vortex for *each* element of $\pi_1^{\mathrm{orb}}(\mathcal{M})$. Moreover, as we have seen in Sec. 2.2.1, $\pi_1^{\mathrm{orb}}(\mathcal{M}) = \Gamma/H$, where $H$ is a normal subgroup of $\Gamma$ whose action fixes $\mathcal{M}$, and it is in fact true that there is a duality vortex associated to each element of $\Gamma$.

What are these duality vortices for Type IIB that we expect from magnetic charge completeness of $\Gamma$? They are nothing other than the familiar 7-branes one finds in F-theory,[9] for reviews see [44, 45]. For instance, recall that the monodromy of a $(p,q)$ 7-brane is specified by an element $T_{p,q} \in \mathrm{SL}(2,\mathbb{Z})$ given by

$$T_{p,q} = \begin{bmatrix} 1 + pq & p^2 \\ -q^2 & 1 - pq \end{bmatrix}. \tag{16}$$

The $T_{p,q}$ all lie in the nontrivial conjugacy class of $\mathrm{SL}(2,\mathbb{Z})$ of matrices $A$ with $\mathrm{tr}\,A = 2$ (excluding the identity). Note that

$$T_{1,0} = \begin{bmatrix} 1 & 1 \\ 0 & 1 \end{bmatrix}, \tag{17}$$

which is precisely the monodromy associated to an ordinary D7-brane. The $(p,q)$ 7-branes therefore form only a single conjugacy class of all duality vortex solutions in Type IIB. All other BPS 7-branes can be formed by placing various combinations of $(p,q)$ 7-branes on top of each other. This supersymmetric operation is sometimes called *fusing* the 7-branes. The resulting 7-branes from the fusion of some set of $(p,q)$ 7-branes will have a monodromy matrix

$$\prod_{k=1}^{n} T_{p_k, q_k}, \tag{18}$$

and one can reach every conjugacy class of $SL(2,\mathbb{Z})$ in this fashion. For example, from the fusion of 4 $T_{1,0}$ branes, one $T_{3,1}$ brane, and one $T_{1,1}$ brane, we form a 7-brane with a monodromy

$$T_{1,0}^4 T_{3,1} T_{1,1} = \begin{bmatrix} -1 & 0 \\ 0 & -1 \end{bmatrix}, \tag{19}$$

which, from the perturbative IIB point of view, is equivalent to four D7-branes coincident with an $O7^-$-plane. Note that this monodromy matrix is the one associated to the center $\mathbb{Z}_2 \in \mathrm{SL}(2,\mathbb{Z})$ that leaves the moduli space unchanged, so it is not possible to draw a closed loop in moduli space associated to this 7-brane. As another example, one can form a 7-brane with a monodromy $S \in SL(2,\mathbb{Z})$ by the fusion $T_{1,0}^6 T_{3,1} T_{1,1}^2$ which has an 8d $\mathfrak{e}_7$ gauge theory localized on its worldvolume (as can be confirmed from studying the $(p,q)$-string junction states [46]).

We are now equipped to study the duality group action on a sector charged objects within the theory – the duality vortices themselves. As we mentioned in section 2.3.1, the charged spectrum of objects of codimension $k > 2$ necessarily enjoys an abelian fusion rule. For the duality vortices of codimension 2, however, the fusion rule satisfied need not be abelian, and indeed, it is simply produced by the nonabelian group multiplication on $\Gamma$.

The reason for this difference can be understood as follows. It is well known [23] that the fusion rule for codimension-1 *genuine* operators is generally nonabelian, while for codimension-2 and higher, the fusion rule is necessarily abelian. However, defects charged under the duality symmetry will generally have "Wilson lines" carrying their discrete 0-form

---

[9]If we more carefully consider the $GL^+$ lift of $SL(2,\mathbb{Z})$ then the additional vortices are the R7 branes of [42,43].

gauge charge, so they must be regarded as non-genuine defects, making the analysis slightly more complicated.[10] Accordingly, suppose we are given two defects $X, Y$ of codimension $k > 2$ and suppose we wish to fuse them in the vicinity of a duality vortex $T_g$ associated to $g \in \Gamma$. If these defects are charged under the duality group in a given representation, then their creation operators are necessarily non-genuine defects similar to how electron creation operators alone are not gauge invariant unless dressed by an attached Wilson line. Therefore $X$ and $Y$ must be understood as the boundaries of defects $\tilde{X}, \tilde{Y}$ of codimension-$(k-1)$. The codimension-$(k-1)$ defects $\tilde{X}, \tilde{Y}$ ending on $X, Y$ can then be understood as fictitious "Wilson lines" carrying away the charge under $\Gamma$ associated to $X, Y$. Thus, the fusion of the defects $X, Y$ is detemined in fact by the fusion of the $\tilde{X}, \tilde{Y}$ in one higher dimension. For $X, Y$ of codimension $k > 2$, the fusion of these defects $\tilde{X}, \tilde{Y}$ is necessarily abelian, since $k - 1 > 1$, while for $X, Y$ of codimension $k = 2$, the fusion rule need not be abelian. Thus, we conclude that the duality vortices have a (possibly) nonabelian fusion rule which is specified by the group $\Gamma$ itself.

With the nonabelian fusion rule of the duality vortices in mind, we can further ask the following: What is the *action* by $\Gamma$ on the duality vortices under monodromy? Consider the depiction in Figure 1. Suppose that we have two vortices with monodromies $M, \tilde{M} \in \Gamma$. Under moving the $M$ vortex around $\tilde{M}$ by $180°$, we have

$$M \mapsto \tilde{M}^{-1} M \tilde{M}. \tag{20}$$

Thus, $\Gamma$ acts on vortices $V(M)$ by the adjoint action, where the monodromy labels the charge of the state. The duality action is therefore specified by the adjoint map

$$\mathrm{ad} : \ \Gamma \rightarrow \mathrm{Aut}(\Gamma), \tag{21}$$

where $\mathrm{Aut}(\Gamma)$ is the automorphism group of $\Gamma$. Moreover, the charge one can associate to a given vortex with monodromy $M$ is given by a conjugacy class $[M] \in \mathrm{Conj}(\Gamma)$ as this data is invariant under braiding with other vortices.

We summarize that the duality action on the theory can be understood in terms of a non-trivial monodromy around a codimension-2 duality vortex in the spacetime picture. These vortices come equipped with a non-abelian fusion rule and transform in the adjoint representation of the duality group. The action of a duality vortex on the spectrum of charged objects is summarized in Figure 1.

We now turn to the case of *instantons*, which are the electric-magnetic duals of duality vortices. Unlike the charged objects discussed in the previous section, instantons are charged under the $(-1)$-form gauge symmetry $\mathcal{M}$ itself as opposed to the higher-form gauge symmetry whose classifying spaces are fibered over $\mathcal{M}$. Now, a fibration over $\mathcal{M}$ is identified naturally with a representation of $\pi_1(\mathcal{M})$ (and therefore the duality group) on the fiber. Instantons, being charged under $\mathcal{M}$ itself, therefore, are not equipped with an action by $\Gamma$ in the literal sense.

Such instantons are ubiquitous in string theory: They can arise already in ten dimensions, like the D$(-1)$-instanton in Type IIB, and they can arise more generally in lower dimensions, where one generically expects Euclidean D-brane instantons to arise from wrapping the entire worldvolume of D-branes on compact cycles. Thus, one may wonder how, if at all, these states are acted on by the duality group and how they might fit in the discussion of the charged spectrum in section 2.3.1.

---

[10] From the perspective of an EFT far below the scale of the 7-brane tension, a duality vortex can be thought of as a non-genuine (in the sense of [47]) codimension-2 Gukov-Witten operator on which a topological codimension-1 operator, which implements the nonabelian duality action, can end. In the deep IR, one can think of these codimension-1 objects as domain walls separating two vacua of the theory, which end on the duality vortex. Alternatively, the domain wall can be seen to correspond to a choice of a branch cut that ends at a branch point corresponding to a duality vortex. (Note that this picture is only meant to be understood topologically; in dynamical settings, the local codimension-1 object can "smear" out into a continuous monodromy action.)

Before we study specific examples, let us briefly discuss what it means to specify the *charge* of an instanton in the first place: It is not entirely clear *a priori* what is meant by a charged object under a $(-1)$-form symmetry. To obtain some intuition for this, let us first review the case of 0-form symmetries. For a continuous, invertible 0-form gauge symmetry, specified for instance by a Lie group $G$ over $\mathbb{C}$, a charged object would be associated to a representation $V$ of $G$ over $\mathbb{C}$. The 0-form gauge symmetry would then associate such a representation $V$ to each dimension-0 submanifold of the ambient bulk.

Since a 0-form invertible gauge symmetry $G$ has charges labeled by the representations of $G$, one now asks what the corresponding notion is for a $(-1)$-form gauge symmetry. For this, we briefly borrow some results from category theory. A representation is understood mathematically as a functor $\rho : BG \to \text{Vect}_{\mathbb{C}}$, where $BG$ is considered as the one-object category whose morphisms present $G$ and $\text{Vect}_{\mathbb{C}}$ is the category of complex vector spaces. But notice that a 0-form gauge symmetry is represented by an entire category $BG$, while a $(-1)$-form gauge symmetry is represented by just a set $S$. Thus, a $(-1)$-form charge should be even simpler than a 0-form charge: For a $(-1)$-form symmetry specified by $S$, we therefore expect a *function $S \to \mathbb{C}$* to be associated to a dimension-0 manifold.

Returning to the case of continuous $(-1)$-form gauge symmetries, we see that each instanton is "charged" with a function $f : \mathcal{M} \to \mathbb{C}$. Physically, this function is precisely the (exponentiated) instanton action. Special properties enjoyed by the instanton actions then encode properties of the corresponding instanton objects. For instance, in supersymmetric examples, a BPS instanton would correspond to a holomorphic function on $\mathcal{M}$.

Now, it becomes clear how the instanton objects "see" the (quotient of the) duality given by $\Gamma' = \pi_1(\mathcal{M})$. For each instanton $f : \mathcal{M} \to \mathbb{C}$, there is a corresponding $\Gamma'$-invariant lift $\hat{f} : \hat{\mathcal{M}} \to \mathbb{C}$ satisfying the following:

$$\hat{f}(x) = \hat{f}(gx), \qquad x \in \mathcal{M}, \quad g \in \Gamma'. \tag{22}$$

Studying the full spectrum of instanton charges $f$ will therefore tell us the duality symmetry acting on $\hat{\mathcal{M}}$, the quotient by which is $\mathcal{M}$.

We now take again the example of Type IIB in ten dimensions to illustrate these claims. One would want to add $D(-1)$ instantons to the list of solitons of Type IIB string theory that are charged under $SL(2, \mathbb{Z})$. Indeed, at least from the perspective of the $D(-1)$ instanton action, given by [48]:

$$S_{D(-1)} = 2\pi\tau, \tag{23}$$

it is clear that the instanton transforms when an element of $SL(2, \mathbb{Z})$ acts on the axio-dilaton $\tau$. Note that the presence of such contributions (23) to the path integral is one of the reasons why the duality group of Type IIB is reduced from $SL(2, \mathbb{R})$ to $SL(2, \mathbb{Z})$. Indeed, the term (23) contributes a factor of $\exp(2\pi i \tau)$ to the path integral, which breaks the continuous shift symmetry of $\tau$ to discrete shifts $\tau \mapsto \tau + n$, $n \in \mathbb{Z}$. Thus, by including the $D(-1)$-instantons in the theory, we conclude that the full duality group must contain as a subgroup the $\mathbb{Z}$ acting by the integer shift which preserves the exponentiated instanton action. This integer $n$ can be interpreted as the $D(-1)$ instanton's magnetic charge under the 9-form flux dual to the axion $C_0$ in ten dimensions. The action of Euclidean instantons in lower dimensional theories can also be shown to generically depend on the moduli, and thus transform under the action of the duality group.

As discussed above, what sets instantons apart from charged states discussed in section 2.3.1, is that they are not directly charged under the various $U(1)$ $p$-form gauge fields with $p \geq 0$ in the Type IIB but instead under the scalars themselves. Indeed, although the higher dimensional branes themselves carry charges under the $U(1)$ $p$-form fields, once such a brane becomes an instanton by wrapping a cycle in the compact space, these charges contribute to

the couplings of the instanton to the various moduli in the EFT. These couplings are encoded in the instanton action, which is precisely the function mentioned above.

Another reason why instantons are not to be put on equal footing with the charged objects discussed above is that instantons are not really objects to begin with: they can simply be considered to be non-perturbative corrections to the effective action. In particular, they lead to corrections to the metric on moduli space, and so they will play a role in our discussion of compactifiability in section 4.3.2.

Finally, the fact that individual instanton actions enjoy are invariant under the action of the duality group (in the sense of Eq. (22)) should not come as a surprise. In fact, the required invariance of the instanton action under duality has been used in the literature to constrain possible instanton corrections to the effective action: Once all of the quantum corrections are incorporated, the full action should be invariant under the duality group. Moreover, by definition, instanton contributions furnish perturbative contributions in dual frame. This is another way in which one can use dualities to determine the exact form of instanton corrections. For example, mirror symmetry (and in particular, the c-map) was used in [49] to compute instanton corrections to the metric on the hypermultiplet moduli space in CY compactifications of type II string theory. This mechanism is precisely what brings an infinite distance boundary of the moduli space to finite distance. This fact will be discussed further in section 4.3.2.

## 2.4 Semisimple representations

We now resume our consideration of duality action on defects of codimension $k > 2$. As we have argued in Sec. 2.3.1, these objects organize themselves as a finitely-generated abelian group, so we wish to understand duality representations on such charge lattices. We provide evidence towards a partial answer to the question posed at the end of section 2.3.1. In particular, we will see from Swampland principles that the charge lattice must be a *semisimple* representation of the duality group $\Gamma$. We will therefore motivate and introduce the notion of semisimplicity and argue for the semisimplicity of duality group representations. In Sec. 2.4.1, we explain and study semisimplicity in the context of *complex* representations of the duality group, obtained by tensoring a free abelian group on which $\Gamma$ acts with $\mathbb{C}$. In Sec. 2.4.2, we briefly analyze the semisimplicity criterion in the case of torsion-valued charges as well.

### 2.4.1 Complex-valued charges

We first turn our attention to the semisimplicity of duality actions on charge lattices $\Lambda_k$. However, as discussed in Sec. 2.3.1, the problem of representation theory becomes substantially easier when working over a field. Thus, we wish to consider instead the action of $\Gamma$ on $\Lambda_k \otimes \mathbb{C}$ (one could instead consider $\Lambda_k \otimes \mathbb{R}$, for which the discussion in this section will remain essentially unchanged).

To begin, let us return to the example of Type IIB in 10d. Note that the action of $SL(2, \mathbb{Z})$ over the complexified vector space $\mathbb{C}^2$ of $(p, q)$-strings and 5-branes is *irreducible*; that is, there is no nontrivial subrepresentation of $SL(2, \mathbb{Z})$. More generally, $SL(2, \mathbb{Z})$ does *not* act irreducibly over the entire charge lattice of Type IIB in 10d; for instance, the D3-brane constitutes a (trivial) subrepresentation of the $SL(2, \mathbb{Z})$ action. However, it *is* true that the representation $V$ of $SL(2, \mathbb{Z})$ on the BPS spectrum (of codimension $k > 2$ states) of the theory decomposes as a direct sum of irreducibles:

$$V = \bigoplus_i V_i, \qquad V_i \text{ is irreducible.} \tag{24}$$

This property enjoyed by $V$ is called *semisimplicity* (or *complete reducibility*). In fact, it is a general property of the group $\Gamma = SL(2, \mathbb{Z})$ that all of its representations are necessarily semisimple. A group with this property is itself called semisimple.

Importantly, semisimplicity is not satisfied by the representations of every naively plausible duality group. For instance, let us consider the upper-triangular subgroup $B \subset \Gamma = \mathrm{SL}(2, \mathbb{Z})$ generated by $\tau \to \tau + 1$, given by

$$B = \left\{ \begin{bmatrix} 1 & n \\ 0 & 1 \end{bmatrix} \middle| n \in \mathbb{Z} \right\}. \tag{25}$$

Note that $B \simeq \mathbb{Z}$ as abelian groups. However, the standard 2-dimensional matrix representation $\mathbb{C}^2$ of $B$ is *not* semisimple: The subspace spanned by $(1, 0)$ is acted on trivially by $B$, but $\mathbb{C}^2$ does not decompose as a direct sum of two one-dimensional representations of $B$. More generally, lattices within upper- or lower-triangular subgroups of semisimple Lie groups will not be semisimple. Returning to the fact that $B$ appears not to be realized in the quantum gravity landscape, it seems therefore that the semisimplicity of $\Gamma$ appears to be a plausible Swampland constraint.

Semisimplicity can be phrased intuitively in terms of a "democratic" action of the duality group on the charge lattice. That is, suppose we are given a duality representation $V$. If there exist subspaces $U, W \subset V$ such that the subspace spanned by $\Gamma \cdot U$ contains $W$, then it should also be true that the subspace spanned by $\Gamma \cdot W$ should contain $U$. Note that this condition is expressly violated by the upper-triangular group $B$ – the subspace spanned by $(1, 0)$ lies in the subspace spanned by action of $\mathrm{SL}(2, \mathbb{Z})$ on $(0, 1)$, but $\mathrm{SL}(2, \mathbb{Z})$ cannot take $(1, 0)$ outside its own subspace.

Let us take a brief moment to introduce an equivalent notion of semisimplicity which motivates its physical relevance. We may directly see that the representation theory of some group $G$ is semisimple over $\mathbb{C}$ iff for complex representations $R, R'$ of $G$, we have

$$R = R' \quad \Longleftrightarrow \quad \mathrm{tr}_R(g) = \mathrm{tr}_{R'}(g), \qquad \forall g \in G. \tag{26}$$

That is, two representations over $\mathbb{C}$ are equivalent iff they have the same trace over $\mathbb{C}$.

In physical terms, this can be understood as follows. For $G$ a Lie group over $\mathbb{C}$, note that the observable associated to any Wilson loop carrying a representation $R$ of $G$ propagating in a spacetime background $X$ is given precisely by

$$W(C) = \mathrm{tr}_R \exp\left( i \oint_C A \right), \tag{27}$$

where $A \in \Omega^1(X, \mathfrak{g})$ is a Lie-algebra valued 1-form for $\mathfrak{g} = \mathrm{Lie}(G)$. In other words, we see that every observable associated to an object in a representation $R$ is given by a trace on $R$. Thus, if the physics is able to distinguish two different objects, they had better have different trace functions! Note that every unitary representation of a compact Lie group is semisimple, so it is in some sense that semisimplicity extends the expected features of unitarity for Lie group symmetry to discrete 0-form duality groups as well.

In Sec. 4, we will provide strong evidence for the semisimplicity property of the duality representation on periods in CY3 compactifications, relating in the process these representation-theoretic statements to the *geometry* of the associated complex structure moduli spaces. In particular, we will see that the semisimplicity of CY3 monodromy groups is indeed true and follows from the *compactifiability* of the associated moduli space, which we will further define and physically motivate in Sec. 5. In Section 4.3, we will describe a series of examples which illustrated duality action on charge lattices. Before that, we will close this section with some remarks on how objects with torsion-valued charges fit in our discussion of charge lattices and semisimplicity.

### 2.4.2 Torsion-valued charges

Above, we argued that charged states transform in semisimple representations of the duality group. However, we have so far only considered integer-valued charges. In this section, we open ourselves to the possibility of objects with torsion-valued charges. Such objects are ubiquitous in string theory, and one can ask how our above discussion can accommodate them.

Recall from Sec. 2.3.1 that the action of a duality group $\Gamma$ on objects of codimension $k > 2$ is that of a group action on a finitely-generated abelian group, of which we considered previously only the free part. The torsion-valued charges of the theory therefore organize into an abelian group Tors, which can be written as

$$\text{Tors} = \bigoplus_{k \geq 1} \mathbb{Z}_k^{n_k} ,$$

where all but finitely many of the $n_k$ are zero.

For the case of duality action on lattices, we investigated the representation theory of the associated complex vector space. However, tensoring with $\mathbb{C}$ annihilates the torsion-valued charges, so this approach cannot be used to understand the action of $\Gamma$ on Tors. In general, even "trivial" group actions on torsion-valued charges will not be semisimple in the naive sense. For instance, consider the action of the multiplicative group $\mathbb{Z}_4^\times \simeq \mathbb{Z}_2$ on $\mathbb{Z}_4$. Then $\mathbb{Z}_4$ can be viewed as a $\mathbb{Z}_2$-module. But $\mathbb{Z}_4$ has a nontrivial submodule $\mathbb{Z}_2 \subset \mathbb{Z}_4$ which does not decompose as a direct summand. From our intuition with vector spaces, we generally expect the multiplicative group to act semisimply, so it appears that the notion of semisimplicity does not appear to make sense when it comes to group actions on general torsion charges.

However, for the case of $\mathbb{Z}_p$-valued charges for $p$ prime, then $\mathbb{Z}_p$ is a *field*, in which case it is possible to talk about vector spaces and representations just like we would with $\mathbb{C}$ and $\mathbb{R}$. We can then study the representation theory of $\Gamma$ over finite fields, and the semisimplicity condition makes good sense. Thus, in analogy with the discussion at the beginning of Sec. 2.4, we propose that the action of $\Gamma$ on $\mathbb{Z}_p^n$ be semisimple for any prime $p$. It is for the case $p = 2$ that we consider a simple example now.

To illustrate all of this, we now consider O3-planes in Type IIB string theory. The space transverse to an O3-plane is known to look like $\mathbb{RP}^5$, which carries non-trivial torsion 3-cycles. This is known to lead to four different O3-plane variants [50]. We now review this construction in more detail. By definition, the O3-plane acts on the NSNS $B_2$-field by a sign reversal, so its corresponding $\mathbb{Z}_2$-valued charge is given by the twisted cohomology class $[dB_2] = [H_3] \in H^3(\mathbb{RP}^5, \widetilde{\mathbb{Z}}) = \mathbb{Z}_2$. Here $\widetilde{\mathbb{Z}}$ denotes the integer coefficient ring twisted with respect to the involution $\iota$ of $S^5$, the quotient by which gives the projective space $\mathbb{RP}^5$[11] via $\mathbb{RP}^5 = S^5/\iota$. The RR 2-form $C_2$ is also twisted under the orientifold and thus takes values $[F_3] \in H^3(\mathbb{RP}^5, \widetilde{\mathbb{Z}}) = \mathbb{Z}_2$. The possibility of switching on off these two different kinds of fluxes leads to four different variants of orientifold planes. The O3$^-$-plane is the variant that carries a trivial charge under both of these torsional classes. The O3$^+$ is the one that carries a non-trivial $\mathbb{Z}_2$-valued charge corresponding to $[dB_2] = [H_3] \in H^3(\mathbb{RP}^5, \widetilde{\mathbb{Z}}) = \mathbb{Z}_2$. On top of that, both variants can acquire a non-trivial $\mathbb{Z}_2$-valued charge corresponding to the RR 2-form given by $[F_3] \in H^3(\mathbb{RP}^5, \widetilde{\mathbb{Z}}) = \mathbb{Z}_2$. We denote by a tilde the variants that carry non-trivial discrete $F_3$-charge.

One way that these torsion-valued charges can be realized is by considering the intersection of an O3-plane along the directions $\{x^0, \ldots, x^2, x^6\}$ with an NS5-brane along $\{x^0, \ldots, x^5\}$ and a D5-brane along $\{x^0, \ldots, x^2, x^7, x^8, x^9\}$ [50]. These branes are fractional in the sense that they are their own image under the orientifold; thus, they cannot move off of it alone. For this reason, these branes effectively carry half-units of full $(p,q)$-fivebrane charge. In this

---

[11]For a review on how to calculate twisted cohomology groups of spheres, see Chapter 3.H of [51].

way, we can build the three other variants from the O3$^-$ by adding in various combinations of "half-branes". We summarize the possible half-brane variants, their torsion-valued charges, and their constructions in terms of intersecting branes in Table 1. We can use this picture to understand how the O3-plane variants transform under the SL(2, $\mathbb{Z}$) transformations.

One can see from the intersecting brane picture or simply from how $(B_2, C_2)$ transform under SL(2, $\mathbb{Z}$) that these four variants are mapped to each other by SL(2, $\mathbb{Z}$) transformations. For instance, the O3$^+$ and the $\widetilde{O3}^-$ map to each other by S-transformation, and the $\widetilde{O3}^-$ and the $\widetilde{O3}^+$ are mapped to each other by a T-transformation.

Motivated by this, we can consider the $\widetilde{O3}^-$ and the O3$^+$ to be a *basis* of "building blocks" over the finite field with two elements, which we denote by $\mathbb{Z}_2$. Note that this mirrors the picture where the D1-brane and the F1-string are a basis of building blocks for $(p, q)$ strings over $\mathbb{Z}$. The O3-planes are then seen to fit in the 2-dimensional defining representation of SL(2, $\mathbb{Z}$) over $\mathbb{Z}_2^2$. The O3$^-$ and the $\widetilde{O3}^+$ variants can be obtained from the building blocks by the action of elements of SL(2, $\mathbb{Z}$). In contrast with the $(p, q)$-strings, there are only a finite number of O3-planes that one can get this way, which is precisely due to the fact that their charges are torsion-valued. In summary, we have shown that all O3-plane variants transform in the defining representation of SL(2, $\mathbb{Z}$), confirming that the action of the duality group on $\mathbb{Z}_2^2$ is semisimple.

## 3 Examples of duality actions

In the previous section we described the action of the duality group on charged objects of codimension greater or equal to two, relying on the SL(2, $\mathbb{Z}$) duality of Type IIB to illustrate our points. We now turn to numerous other examples of duality groups in string theory to provide further evidence for our claims.

### 3.1 Symmetric moduli spaces

We first discuss the example of *symmetric* moduli spaces, which arise in theories with at least 16 supercharges. As a first example, consider M-theory on $T^n$, which preserves all 32 supercharges of the 11d theory. The associated moduli space is a double quotient which is acted on by the $U$-duality group $E_{n(n)}(\mathbb{Z})$, a lattice within the split real form $E_{n(n)}$ of $E_n$. The full moduli space is

$$\mathcal{M} = E_{n(n)}(\mathbb{Z}) \backslash E_{n(n)} / K_n \,, \tag{28}$$

where $K_n$ is the maximal compact subgroup of $E_{n(n)}$. As discussed in [26], the marked moduli space is given by $\hat{\mathcal{M}} = E_{n(n)} / K_n$, which is a (contractible) Riemannian symmetric space of noncompact type, and the duality group is $\Gamma = E_{n(n)}(\mathbb{Z})$.

Table 1: Four variants of $O_p$-planes, their torsion-valued charges, and their relation to the O3$^-$ by the addition of various "half-branes".

| $(H_3, F_{6-p})$ | type | Intersecting brane picture |
|:---:|:---:|:---:|
| $(0, 0)$ | O3$^-$ | $O3^-$ |
| $(\frac{1}{2}, 0)$ | O3$^+$ | $O3^- + \frac{1}{2}NS5$ |
| $(0, \frac{1}{2})$ | $\widetilde{O3}^-$ | $O3^- + \frac{1}{2}D5$ |
| $(\frac{1}{2}, \frac{1}{2})$ | $\widetilde{O3}^+$ | $O3^- + \frac{1}{2}NS5 + \frac{1}{2}D5$ |

By magnetic charge completeness, we again expect codimension-2 defects for M-theory compactified on $T^n$ associated to every element of $E_{n(n)}(\mathbb{Z})$. Elements $g \in E_{n(n)}(\mathbb{Z})$ which have a trivial image in the dimension-1 cobordism group $\Omega_1(BE_{n(n)}(\mathbb{Z})) \simeq \mathrm{Ab}(E_{n(n)}(\mathbb{Z}))$, where Ab denotes abelianization, can be associated with gravitational solitons [38].

What is the charge lattice of this theory on which $\Gamma$ acts? This question is a bit subtle to answer for the *entire* $\Gamma$ due to the non-geometric and nonperturbative nature of the full $U$-duality. Nevertheless, we can study what happens for various subgroups of the $U$-duality group which can be understood geometrically in different dual descriptions. For a simple example, take M-theory on $T^3$. Then the full $U$-duality group is given by

$$\Gamma = E_{3(3)}(\mathbb{Z}) = \mathrm{SL}(3,\mathbb{Z}) \times \mathrm{SL}(2,\mathbb{Z}). \tag{29}$$

The $\mathrm{SL}(3,\mathbb{Z})$ factor can be realized as the group of large diffeomorphisms of $T^3$. Indeed, $H_1(T^3) = H_2(T^3) = \mathbb{Z}^3$, on which $\mathrm{SL}(3,\mathbb{Z})$ acts by outer automorphism. The charged spectrum of the 8d theory then consists of strings/particles charged electrically under the $C_3$-field wrapped around nontrivial 1-cycles/2-cycles in $T^3$, as well as 4-branes/3-branes electrically charged under the dual $C_6$ field wrapped around nontrivial 1-cycles/2-cycles in $T^3$. These states of various dimensions each parametrize $\mathbb{Z}^3$ lattices. The $\mathrm{SL}(2,\mathbb{Z})$ factor cannot be seen geometrically from the M-theory side, but it can be understood easily in terms of the dual F-theory description.[12] Indeed, the $\mathrm{SL}(2,\mathbb{Z})$ acts on the F-theory elliptic fiber and in turn the Type IIB axiodilaton, $(p,q)$-strings, and $(p,q)$-fivebranes. A reduction of Type IIB on $T^2$ then preserves this $\mathbb{Z}^2$ charge lattice in 8d on which the $\mathrm{SL}(2,\mathbb{Z})$ factor of $\Gamma$ acts. Finally, note that the charged vortices associated with a given $g \in SL(3,\mathbb{Z})$ can be realized by gravitational solitons because $\mathrm{Ab}(SL(3,\mathbb{Z})) = 0$. For more general computations of bordism groups relevant for the U-duality group (29), see [52].

Let us now mention the case of theories with 16 supercharges. In this case, the vector multiplet moduli space is the Narain moduli space, given by a double quotient of the following form:

$$\mathcal{M} = \mathrm{SO}(n,r;\mathbb{Z}) \backslash \mathrm{SO}(n,r)/(\mathrm{SO}(n) \times \mathrm{SO}(r)). \tag{30}$$

In this case, $\hat{\mathcal{M}} = \mathrm{SO}(n,r)/(\mathrm{SO}(n) \times \mathrm{SO}(r))$, is also a contractible symmetric space of non-compact type. The duality group is $\Gamma = \mathrm{SO}(n,r;\mathbb{Z})$. The charge lattice in this example is then parametrized by the electrically and magnetically charged objects which couple to the $U(1)^{n+r}$ 1-form gauge fields, which enjoys the natural action of $\mathrm{SO}(n,r;\mathbb{Z})$. For these cases, it would be interesting to study the corresponding vortices required by completeness of the spectrum of magnetic states.

## 3.2  4d $\mathcal{N} = 2$ vector multiplet moduli spaces

We now turn our attention to more nontrivial examples of moduli spaces in quantum gravity. In order to illustrate the above discussion in a more involved setting, we turn to Calabi-Yau threefold (CY3) compactifications of Type IIB string theory, resulting in 4d $\mathcal{N} = 2$ supergravity theories. These theories can be divided into a vector multiplet sector and a hypermultiplet sector, which we will discuss in turn. The main subject of this section will be the vector multiplet sector, while we defer the discussion of the hypermultiplet sector to Sec. 3.3. Infinite distance limits in these vector multiplet moduli spaces have been classified based on monodromies [53–56] and the emergent string conjecture [57]. In the following we will lay out the mathematical and physical properties underlying them.

Let us first review some generalities about the complex structure moduli space of Calabi–Yau manifolds. The monodromy group $\Gamma$ arises from a variation of Hodge structure over the

---

[12]Recall that M-theory on a finite size $T^3$ is equivalent to F-theory on $T^3 \times S^1$ where a $T^2 \subset T^3$ is identified with the elliptic fiber and the length of the $S^1$ is proportional to the area of $T^2 \subset T^3$ in the M-theory duality frame.

complex structure moduli space $\mathcal{M}_{cs}$. Namely, the middle cohomology $H^3(X, \mathbb{C})$ of a CY3 $X$ can be viewed as a $(2h^{2,1} + 2)$-dimensional vector space, which admits a Hodge decomposition into the subspaces $H^{p,q}$ of $(p, q)$-forms. For instance, take the unique holomorphic $(3, 0)$-form $\Omega$, and consider its period vector

$$\Pi^I(z) = \int_{\Gamma_I} \Omega(z), \tag{31}$$

where $\Gamma_I \in H_3(X, \mathbb{Z})$ denotes an integral basis of 3-cycles. As we move through the moduli space, the period vector $\mathbf{\Pi}(z)$ varies. In particular, winding around a singularity, say at $z^i = 0$, induces a monodromy transformation

$$z^i \mapsto e^{2\pi i} z^i : \qquad \mathbf{\Pi}(z) \mapsto M \cdot \mathbf{\Pi}(z), \tag{32}$$

where $M \in Sp(2h^{2,1} + 2, \mathbb{Z})$. The monodromies induced by all loops $\pi_1(\mathcal{M}_{cs})$ generate the monodromy group $\Gamma \subseteq Sp(2h^{2,1} + 2, \mathbb{Z})$. Furthermore, as we discuss in more detail in section 4, $\Gamma$ must act semisimply. The monodromy matrices around the singular loci are not given by arbitrary symplectic matrices; indeed, they must additionally be *quasi-unipotent* [58]. That is, the matrices $M$ must satisfy the following conditions:

$$(M^l - 1)^k \neq 0, \qquad (M^l - 1)^{k+1} = 0, \tag{33}$$

for some integers $k, l \in \mathbb{N}$.[13] Considering the Jordan decomposition of $M$, we then find from (33) that the semisimple and unipotent factors must obey

$$M = M_{ss} M_u, \qquad (M_{ss})^l = 1, \qquad M_u = e^N, \qquad N^k \neq 0, \qquad N^{k+1} = 0. \tag{34}$$

The behavior of the periods close to a given boundary can be described precisely through these monodromy matrices, known as the nilpotent orbit approximation (referring to the nilpotent matrix $N$ above); see [53–56] for applications in the context of the distance conjecture.

Having covered the main general properties of Calabi–Yau monodromies, let us now turn to an explicit set of examples. Concretely, we consider Calabi–Yau manifolds with $h^{2,1} = 1$ whose complex structure moduli space is given by the thrice-punctured sphere $\mathcal{M}_{cs} = \mathbb{P}^1 - \{0, 1, \infty\}$. There are 14 such Calabi–Yau threefolds [59, 60], including for example the mirror quintic [61]. In fact, all mirrors of these Calabi–Yau manifolds are realized as complete intersections in weighted projective spaces. All these examples are summarized in table 2. For studies of the distance conjecture in these examples we refer to [62–64].

We can specify the monodromy matrices in terms of the topological data of the Calabi–Yau threefold $Y$ mirror dual to $X$: the intersection number $\kappa$ and integrated second Chern class $c_2$. The large complex structure monodromy around $z = \infty$ reads

$$M_\infty = \begin{bmatrix} 1 & 0 & 0 & 0 \\ 1 & 1 & 0 & 0 \\ -\frac{\kappa}{2} - \sigma & -\kappa & 1 & 0 \\ \frac{c_2 + 2\kappa}{12} & \frac{\kappa}{2} - \sigma & -1 & 1 \end{bmatrix}, \tag{35}$$

where $\sigma = \kappa/2 \mod 1$. At the same time, the conifold monodromy around $z = 1$ is given by

$$M_1 = \begin{bmatrix} 1 & 0 & 0 & -1 \\ 0 & 1 & 0 & 0 \\ 0 & 0 & 1 & 0 \\ 0 & 0 & 0 & 1 \end{bmatrix}. \tag{36}$$

---

[13]More precisely, the nilpotency degree is bounded by the dimension of the CY3 as $k \leq 3$, while the order is bounded through the Euler totient function as $\phi(l) < 2h^{2,1} + 2$.

Here we can already see the semisimple action of the monodromy group via a single irreducible representation. Namely, we have brought the large complex structure monodromy $M_\infty$ into a lower-triangular form where $N_\infty = \log M_\infty$ has nilpotency degree $k = 3$. On the other hand, the conifold monodromy $M_1$ acts precisely on the highest-weight state, $(1, 0, 0, 0)$, of $N_\infty$ by shifting it by the lowest-weight state $(0, 0, 0, 1)$.

Let us now turn to the singularity types that can arise at the remaining point $z = 0$. It is customary to characterize these points through the local exponents $0 < a_1 \leq a_2 \leq a_3 \leq a_4 < 1$ of the periods. For the monodromy matrix $M_0$ these exponents correspond to its eigenvalues $e^{2\pi i a_1}, e^{2\pi i a_2}, e^{2\pi i a_3}, e^{2\pi i a_4}$. The singularity types we can encounter are then given by

- Landau-Ginzburg point: $a_1 \neq a_2 \neq a_3 \neq a_4$. This point lies at finite distance, there are no massless states, and has a finite order monodromy. An examples is the $\mathbb{Z}_5$ orbifold point in the moduli space of the mirror quintic.

- Conifold (C) point: $a_1 \neq a_2 = a_3 \neq a_4$. This is a finite distance point but with infinite order monodromy, just like the other conifold point at $z = 1$. There is a single BPS state that becomes massless.

- K3 (K) point: $a_1 = a_2 \neq a_3 = a_4$. This point lies at infinite distance and has an infinite order monodromy. The infinite distance arises because of an emergent string which becomes tensionless.

- Large complex structure (LCS) point: $a_1 = a_2 = a_3 = a_4$. This monodromy is of infinite order and so-called 'maximally unipotent', lying at infinite distance. There is a tower of BPS states that becomes massless arising from (multi-)wrappings of the BPS state $(1, 0, 0, 0)$;[14] this signals a decompactification to 5d M-theory.

Given all of these types of monodromies, it is now a natural question to ask about the full group $\Gamma$ that they generate. Generically, these monodromies do not generate all of the symplectic group but only a subgroup thereof: $\Gamma \subseteq \mathrm{Sp}(4, \mathbb{Z})$. For seven out of the fourteen hypergeometric cases, it was shown in [65] that the monodromy groups are (possibly amalgamated) free products, as summarized in table 2. These monodromy groups are generated by the monodromies around the conifold point $z = 1$ and (antipode of) the LCS point $z = 0$ as

$$\Gamma = \begin{cases} \langle M_1 \rangle * \langle M_0 \rangle, & \text{if } (M_0)^k \neq -1 \text{ for any } k \in \mathbb{N}. \\ \langle M_1, -1 \rangle *_{\mathbb{Z}_2} \langle M_0 \rangle, & \text{if } (M_0)^k = -1 \text{ for some } k \in \mathbb{N}. \end{cases} \tag{37}$$

The conifold monodromy always generates $\langle M_1 \rangle = \mathbb{Z}$, but $\langle M_0 \rangle$ depends on the type of monodromy at $z = 0$. If $M_0$ corresponds to an infinite order monodromy or a Landau-Ginzburg point of odd order $k$ we find respectively that

$$\Gamma = \mathbb{Z} * \mathbb{Z}, \quad \text{or} \quad \Gamma = \mathbb{Z} * \mathbb{Z}_k, \tag{38}$$

while if $z = 0$ is a Landau-Ginzburg point of even order $2k$ we have

$$\Gamma = (\mathbb{Z} \times \mathbb{Z}_2) *_{\mathbb{Z}_2} \mathbb{Z}_{2k}, \tag{39}$$

where the amalgamation is along the relation $(M_0)^k = -1$. For the other seven cases, further obstructions (e.g. additional relations among the generators) prevented a similarly simple characterization of the mondoromy groups. It was argued in [66, 67] that these groups are of finite index in $\mathrm{Sp}(4, \mathbb{Z})$, in contrast to the amalgamated free product groups given above.

---

[14]From the mirror dual Type IIA perspective these states correspond to D0-branes that become massless in Planck units in the large volume limit.

Table 2: Summary of monodromy groups of (mirrors of) complete intersection Calabi-Yau threefolds in weighted projective spaces. In the references [65,68] these cases are labeled by topological data as $d = \kappa$ and $k = \kappa/6 + c_2/12$. The computation of the volumes of the moduli spaces is carried out in section 4.3.2.

| Mirror | $(\kappa, c_2)$ | Exponents | Type | Monodromy group | index | $\mathrm{Vol}(\mathcal{M})$ |
|---|---|---|---|---|---|---|
| $\mathbb{P}_5[1^5]$ | $(5,5)$ | $(\frac{1}{5}, \frac{2}{5}, \frac{3}{5}, \frac{4}{5})$ | LG | $\mathbb{Z} * \mathbb{Z}_5$ | $\infty$ | $2\pi/5$ |
| $\mathbb{P}_8[1^4, 4]$ | $(2,44)$ | $(\frac{1}{8}, \frac{3}{8}, \frac{5}{8}, \frac{7}{8})$ | LG | $(\mathbb{Z} \times \mathbb{Z}_2) *_{\mathbb{Z}_2} \mathbb{Z}_8$ | $\infty$ | $\pi/4$ |
| $\mathbb{P}_{12,2}[1^4, 4, 6]$ | $(1,46)$ | $(\frac{1}{12}, \frac{5}{12}, \frac{7}{12}, \frac{11}{12})$ | LG | $(\mathbb{Z} \times \mathbb{Z}_2) *_{\mathbb{Z}_2} \mathbb{Z}_{12}$ | $\infty$ | $\pi/6$ |
| $\mathbb{P}_{2,2,2,2}[1^8]$ | $(16,64)$ | $(\frac{1}{2}, \frac{1}{2}, \frac{1}{2}, \frac{1}{2})$ | LCS | $\mathbb{Z} * \mathbb{Z}$ | $\infty$ | $\pi$ |
| $\mathbb{P}_{3,2,2}[1^7]$ | $(12,60)$ | $(\frac{1}{3}, \frac{1}{2}, \frac{1}{2}, \frac{2}{3})$ | C | $\mathbb{Z} * \mathbb{Z}$ | $\infty$ | $2\pi/3$ |
| $\mathbb{P}_{4,2}[1^6]$ | $(8,56)$ | $(\frac{1}{4}, \frac{1}{2}, \frac{1}{2}, \frac{3}{4})$ | C | $\mathbb{Z} * \mathbb{Z}$ | $\infty$ | $\pi/2$ |
| $\mathbb{P}_{6,2}[1^5, 3]$ | $(4,52)$ | $(\frac{1}{6}, \frac{1}{2}, \frac{1}{2}, \frac{5}{6})$ | C | $\mathbb{Z} * \mathbb{Z}$ | $\infty$ | $\pi/3$ |
| $\mathbb{P}_{10}[1^3, 2, 5]$ | $(1,34)$ | $(\frac{1}{10}, \frac{3}{10}, \frac{7}{10}, \frac{9}{10})$ | LG | [68] | $6$ | $\pi/5$ |
| $\mathbb{P}_{6,6}[1^2, 2^2, 3]$ | $(1,22)$ | $(\frac{1}{6}, \frac{1}{6}, \frac{5}{6}, \frac{5}{6})$ | K | [68] | $10$ | $\pi/3$ |
| $\mathbb{P}_{6,4}[1^3, 2^2, 3]$ | $(2,32)$ | $(\frac{1}{6}, \frac{1}{4}, \frac{3}{4}, \frac{5}{6})$ | LG | | $960$ | $\pi/3$ |
| $\mathbb{P}_6[1^4, 2]$ | $(3,42)$ | $(\frac{1}{6}, \frac{1}{3}, \frac{2}{3}, \frac{5}{6})$ | LG | | $2^9 3^5 5^2$ | $\pi/3$ |
| $\mathbb{P}_{4,4}[1^4, 2^2]$ | $(4,40)$ | $(\frac{1}{4}, \frac{1}{4}, \frac{3}{4}, \frac{3}{4})$ | K | | $2^{20} 3^2 5$ | $\pi/2$ |
| $\mathbb{P}_{4,3}[1^5, 2]$ | $(6,48)$ | $(\frac{1}{4}, \frac{1}{3}, \frac{2}{3}, \frac{3}{4})$ | LG | | $\geq 2^{10} 3^6 5^2$ | $\pi/2$ |
| $\mathbb{P}_{3,3}[1^6]$ | $(9,54)$ | $(\frac{1}{3}, \frac{1}{3}, \frac{2}{3}, \frac{2}{3})$ | K | | $\geq 2^8 3^{13} 5$ | $2\pi/3$ |

See (40) for a further discussion. In [68] two of these seven remaining groups were explicitly described as finite index subgroups of $\mathrm{Sp}(4, \mathbb{Z})$. The corresponding indices were determined in three more cases, and lower bounds were placed on the index in the two remaining cases. A summary of these results is given in 2.

Given this landscape of monodromy groups, let us now draw some general lessons about the possible types of duality groups and their associated generators. We start by comparing the monodromy groups to the familiar example of $\mathrm{PSL}(2, \mathbb{Z})$ from 10d Type IIB string theory. Recall from (9) that it is freely generated by $\mathbb{Z}_2$ and $\mathbb{Z}_3$, corresponding to the two orbifold points at $\tau = i$ and $\tau = e^{2\pi i/3}$. This appears to suggest some plausible properties of duality groups that turn out to be false in general:

- **The duality group is not necessarily generated by finite order elements.** Namely, the free amalgamated monodromy groups $\Gamma$ in table 2 all contain an infinite order factor $\mathbb{Z}$ corresponding to the conifold monodromy (36).

- **The duality group need not even be generated by monodromies around finite distance points.** Namely, the mirror complete intersection $\mathbb{P}_{2,2,2,2}[1^8]$ has monodromy group $\Gamma = \mathbb{Z} * \mathbb{Z}$, and in this case the second free factor $\mathbb{Z}$ is generated by a large complex structure (maximally unipotent monodromy) point at infinite distance.

Since generically monodromy groups are strict subgroups $\Gamma \subseteq \mathrm{Sp}(4,\mathbb{Z})$, one might also wonder whether they must have a finite index. The seven free amalgamated $\Gamma$ give immediate counter-examples, as [65][15]

$$|\mathrm{Sp}(4,\mathbb{Z}):\Gamma| = \infty\,. \tag{40}$$

The other 7 cases do, however, have finite index, as computed in [68] (and given in Table 2 for completeness). In fact, a particular period system (outside of the 14 hypergeometric cases) was identified for which one would obtain $\Gamma = Sp(4,\mathbb{Z})$, although no corresponding Calabi–Yau geometry has been identified for this period system as of the writing of this paper.

We briefly note that there is another completeness criterion for the monodromy group that *is* satisfied in all these examples. We recall that the so-called *Zariski closure* of $\Gamma$, is defined to be the smallest algebraic group over $\mathbb{Z}$ that contains $\Gamma$. Then, letting Zcl($S$) be the Zariski closure of a subgroup $S$ in some larger group, we have

$$\mathrm{Zcl}(\Gamma) = Sp(4,\mathbb{Z})\,, \tag{41}$$

for all 14 hypergeometric monodromy groups [69]. We leave the physical interpretation of this completeness condition for future work.

Having described the moduli space and the duality group, we now turn to the spectrum of states and their duality transformations. Let us begin with the particle states, which are given by D-branes wrapping cycles of the Calabi–Yau manifold. On the Type IIB side, particle states arise from D3-branes wrapping a particular class of 3-cycles $q \in H^3(X,\mathbb{Z})$, while on the Type IIA side, the dual particle states arise as bound states $q = (q_{D0}, q_{D2}, q_{D4}, q_{D6})^T$ of D0, D2, D4, and D6-branes wrapped on 0, 2, 4, and 6-cycles of the mirror $Y$, respectively. From the perspective of the 4d $\mathcal{N} = 2$ supergravity theory we can understand these charges as the electric and magnetic charges under the two $U(1)$ gauge fields in the vector and gravity multiplet. The mass of a given BPS state follows by definition from its central charge $M(q) = |Z(q)|$, which is given by

$$Z(q) = e^{K/2}(q,\Pi)\,. \tag{42}$$

Circling a singularity in the moduli space by $z \to e^{2\pi i}z$, these BPS states transform under the monodromy transformations via

$$\begin{bmatrix} q_{D0} \\ q_{D2} \\ q_{D4} \\ q_{D6} \end{bmatrix} \to M \cdot \begin{bmatrix} q_{D0} \\ q_{D2} \\ q_{D4} \\ q_{D6} \end{bmatrix}\,. \tag{43}$$

This transformation rule is such that the central charge as in Eq. (42) remains invariant under monodromies, since the period vector transforms in the same way (32), and the monodromies are automorphisms of the bilinear pairing with which the BPS spectrum is equipped (arising, for instance, from the skew-symmetric pairing on the middle-dimensional forms of the CY3 in the Type IIB picture). These BPS states thus furnish a semisimple representation of the electromagnetic duality group $\Gamma \subseteq Sp(2h^{2,1}+2;\mathbb{Z})$ induced by the monodromy matrices.

Having discussed the BPS particles, we next identify the duality vortices and their action on the charged spectrum of the theory. The duality vortices must be strings (of codimension two) that implement a duality transformation via monodromy in four dimensions. Let us focus first on the duality vortex associated to the large complex structure monodromy (35). On the Type IIB side this strings is expected to come from 10d Kaluza-Klein monopoles wrapped on a 3-cycle of $X$, while on the Type IIA side they arise from NS5-branes wrapping a divisor of the mirror $Y$. The 4d objects implementing the integer shift 0-form symmetry are nothing but axion strings, see [39, 70–75] for recent studies in light of the swampland program. The axion

---

[15]This infinite index follows from the fact that $Sp(4,\mathbb{Q})$ and all of its finite-index subgroups have cohomological dimension two, while (possibly amalgamated) free product groups have cohomological dimension one.

that winds around this string corresponds to the phase of the complex structure coordinate $z$ in the Type IIB picture, while in the mirror dual Type IIA picture this axion is obtained from dimensionally reducing the NS-NS $B$-field along a 2-cycle.

What about the strings that correspond to monodromies whose fixed points lie in the bulk of moduli space, say the conifold point? Although these objects have appeared sporadically in various corners of the literature (for example, see [39]), they are significantly more mysterious. In order to understand these strings better, it is instructive to compare them to the 7-branes in 10d Type IIB. Recall from section 2.3.2 that we can generate all duality vortices of SL$(2, \mathbb{Z})$, including those of the $\mathbb{Z}_4$ and $\mathbb{Z}_6$ fixed points, from (the duality orbit of) the D7-brane. The way this works is to first consider the D7-brane in all possible duality frames, resulting in the familiar $[p, q]$ 7-branes. Subsequently, we may fuse different types of $[p, q]$ 7-branes together, and we can thereby generate in this way a 7-brane for every possible element of SL$(2, \mathbb{Z})$. We thus find that all 7-branes can in a sense be constructed from a single fundamental building block: the D7-brane.

It is thus natural to ask whether a similar picture holds for the axionic strings considered here, by considering, for instance, the LCS string as a fundamental building block. To be more precise, is it possible to generate the full axionic string spectrum by consider the LCS string across different duality frames and fusing these together? As it turns out, in the case of the (amalgamated) free product groups encountered in Table 2, we find a counterexample. Namely, say that there is some fusion of LCS strings across different duality frames that produces the conifold string. At the level of group elements, this would then amount to a relation of the following form:

$$g_1 M_\infty g_1^{-1} \ldots g_n M_\infty g_n^{-1} = M_1 \,, \tag{44}$$

for some $g_1, \ldots, g_n \in \Gamma$. We can then expand $M_\infty = M_0 M_1$, writing all of the elements $g_1, \ldots, g_n$ out as words in $M_0$ and $M_1$, and reducing by the relations $M_0 M_0^{-1} = 1$ and $M_1 M_1^{-1} = 1$ wherever possible. Then, we find that (44) constitutes a relation between $M_0$ and $M_1$ incompatible with the free structure of the groups given in table 2, furnishing a contradiction.[16] This means we cannot generate all duality vortices of $\Gamma \subset \text{Sp}(4, \mathbb{Z})$ by considering the LCS string across different duality frames and fusing them together. In other words, we need another axionic string as a second fundamental building block. It would be interesting to identify what is the higher-dimensional origin of this axionic string, analogous to how the LCS string comes from an NS5-brane wrapped on a divisor in the mirror dual Type IIA picture. More specifically, if we look at the 7 cases with a free monodromy group in table 2, we see that for 6 out of those an axionic string associated with the Landau-Ginzburg or conifold point is needed. The only exception is $\mathbb{P}_{2,2,2,2}[1^8]$ since it has a second LCS point, so we can consider a second axionic string coming from an NS5-brane wrapping a divisor in the mirror dual picture.[17]

We can also study domain walls (and the associated duality actions) in CY compactifications, provided we consider $\mathcal{N} = 1$ orientifold compactifications of Type IIB rather than the $\mathcal{N} = 2$ geometric examples considered thus far. In these examples, domain walls correspond to three-form RR and NS-NS fluxes $F_3, H_3 \in H^3(X, \mathbb{Z})$.[18] These fluxes induce a superpoten-

---

[16]Note that the same argument does not apply to $SL(2, \mathbb{Z}) = \mathbb{Z}_4 *_{\mathbb{Z}_2} \mathbb{Z}_6$, where such relations are possible. There the caveat in this case is that the D7-brane corresponds to $T = S^{-1}U$, where $S$ and $U$ are the generators of $\mathbb{Z}_4$ and $\mathbb{Z}_6$, so it is given by the product of two finite-order generators instead. The corresponding fusion relations are well-known [44, 45], see for example Eq. (19).

[17]Mirror symmetry predicts that for each LCS point in the complex structure moduli space of a CY3 $X$ there is a corresponding mirror dual. In this way the two LCS points provide us with two mirrors—one of which is $\mathbb{P}_{2,2,2,2}[1^8]$—and for each we get an axionic string from wrapping an NS5-brane on its divisor.

[18]In general, we should consider the action of the holomorphic involution of the CY3 $X$ on the three-form cohomology [76]. For simplicity we take all of the three-form cohomology to be odd under this involution, so that we can ignore these subtleties.

tial [77] given by

$$W(G_3, t) = \int_X G_3 \wedge \Omega(t) = (G_3, \Pi(t)),\qquad(45)$$

where we have defined the complex three-form flux $G_3 = F_3 - \tau H_3$ with $\tau$ the axiodilaton of Type IIB. Solutions to the F-terms equations $D_I W = 0$ for the moduli correspond to minima of the corresponding scalar potential. Finiteness of the number of these flux vacua is proven in [78,79] on general grounds for F-theory Calabi–Yau fourfold compactifications by using the tadpole bound. Note here that different choices of fluxes may be identified by monodromies: under a monodromy action (32) on the complex structure moduli (say, $t \mapsto t + 1$) we find that the superpotential transforms as

$$W(G_3, t) \to W(G_3, t+1) = (G_3, M\Pi(t)) = (M^{-1}G_3, \Pi(t)),\qquad(46)$$

where we have used the fact that the monodromies are automorphisms of the Hodge star pairing on the three-forms of the CY3 $X$. Thus, we find that the superpotential remains invariant if we simultaneously transform the fluxes as $G_3 \to M G_3$. From the 4d point of view, the fluxes can be viewed as four-form field strengths of non-dynamical three-form gauge potentials. The mixing of this 2-form symmetry with the duality symmetries can then be understood, in the language of generalized symmetries, as an $n$-group symmetry. See [80] for a general review of $n$-group symmetries.

## 3.3 4d $\mathcal{N} = 2$ hypermultiplet moduli spaces

The vector multiplet moduli discussed thus far are fairly well-understood. On the other hand, 4d $\mathcal{N} = 2$ supergravity theories (obtained, for instance, from Type II CY3 compactifications) also possess a hypermultiplet sector, whose moduli are much more mysterious. For instance, consider a case of a compactification of Type IIB on a CY3 $X$ with $h^{1,1} = 1$ (for concreteness, one can take $X$ to be the quintic). The hypermultiplet moduli space of the 4d $\mathcal{N} = 2$ theory is then a quaternionic Kähler manifold in 8 real dimensions fibered over the complex structure moduli space of the mirror quintic $X^\vee$. The moduli space is given by the axio-dilaton $\tau = c_0 + ie^{-\phi}$; another complex scalar $\psi = b_2 + ir$, where $r$ corresponds to the volume of the CY and $b_2$ is the NSNS axion (obtained from dimensionally reducing $B_2$ on the 2-cycle); and four extra axions: three, $c_2, c_4, c_6$, from the RR fields wrapping cycles of the appropriate dimension, and one, $b_6$, from the NSNS 6-form wrapping the entire CY3.

From this picture, it is already clear that the duality group admits several canonical actions. First of all, the monodromy group of the mirror quintic $\Gamma_Q \subset \mathrm{Sp}(4, \mathbb{Z})$ acts non-linearly on the modulus $\psi$ and linearly on the fibered RR axions $c_0, c_2, c_4, c_6$:

$$\psi \mapsto g\psi, \qquad \begin{bmatrix} c_0 \\ c_2 \\ c_4 \\ c_6 \end{bmatrix} \mapsto M_g \begin{bmatrix} c_0 \\ c_2 \\ c_4 \\ c_6 \end{bmatrix}, \qquad g \in \Gamma_Q,\qquad(47)$$

where $M_g$ is the 4-dimensional symplectic matrix representing the transformation $g$. Second, the S-duality of the 10-dimensional Type IIB theory acts nonlinearly on $\tau$, and linearly on the NSNS and RR fluxes:

$$\tau \mapsto \frac{a\tau + b}{c\tau + d}, \qquad \begin{bmatrix} b_2 \\ c_2 \end{bmatrix} \mapsto \begin{bmatrix} a & b \\ c & d \end{bmatrix} \begin{bmatrix} b_2 \\ c_2 \end{bmatrix}, \qquad \begin{bmatrix} b_6 \\ c_6 \end{bmatrix} \mapsto \begin{bmatrix} a & b \\ c & d \end{bmatrix} \begin{bmatrix} b_6 \\ c_6 \end{bmatrix}.\qquad(48)$$

We now wish to understand the charged spectrum of the theory. To do this, we consider higher-dimensional branes wrapping cycles in the CY3, which will generally pick up a monodromy action under the discrete axion shift symmetry. However, we notice that we only have

Table 3: Axions in the hypermultiplet moduli space and the corresponding (wrapped) D-branes and NS5-branes in Type IIA.

| Axion | $c_0$ | $c_2$ | $c_4$ | $c_6$ | $b_2$ | $b_6$ |
|-------|-------|-------------|-------------|-----------|-------|--------------|
| String | D1 | D3 on 2-cycle | D5 on 4-cycle | D7 on CY3 | F1 | NS5 on 4-cycle |

*even*-dimensional branes to wrap around cycles. As such, the only observables in the lower-dimensional theory are coupled to 0, 2, and 4-form fields. The 0-form fields are precisely the axions we have seen previously. The 4-form fields correspond to space-filling operators, which could be considered to contribute to discrete $\theta$-angles of the lower-dimensional EFT. However, they will play no role in understanding the charged "objects" of the EFT on which the duality group acts. Thus, the only available objects are charged under 2-form fields. In 4 dimensions, these are magnetically dual to the axions. This suggests to us more generally that dimension-2 charged objects correspond to vortices characterized by $\pi_1(\mathcal{M})$.

This picture can be made explicit in the context of the higher-form gauge fields in the theory. Consider, for the sake of concreteness, $C_2$ wrapped on a 2-cycle $\gamma$. What is the dual axionic string? It must correspond to the dual D5-brane wrapped on the dual 4-cycle $\gamma^\vee$, which is indeed in the spectrum of our theory. In total we get the spectrum as indicated in Table 3.

We therefore see that the *entire* BPS spectrum acted on by the duality group which acts also on the hypermultiplet moduli consists of the duality vortices themselves. There is no charge lattice to speak of! Thus, letting the full monodromy group be $\pi_1(\mathcal{M}) = \Gamma$, we see that the marked moduli space $\hat{\mathcal{M}}$ must be realizable as the covering space for which the group $\Gamma$ itself is the (discrete) fiber and $\mathcal{M}$ is the base. Recall from Sec. 2.3.2 that the action of duality vortices on themselves is given by the adjoint map:

$$\mathrm{ad}_g : h \mapsto ghg^{-1}. \tag{49}$$

Since the marked moduli space conjecture proposes that the duality vortices are sufficient to mark the moduli space, the adjoint action must be faithful. Recalling that the kernel of the adjoint map $\mathrm{ad} : \Gamma \to \mathrm{Aut}(\Gamma)$ is given by the center $Z(\Gamma)$, we therefore conjecture that

$$Z(\Gamma) = 0. \tag{50}$$

It would be interesting to further study the global topology of the hypermultiplet moduli space to check this conjecture.

Let us now restrict our attention to the fundamental hypermultiplet moduli space, which is spanned by the axio-dilaton $\tau$ and the axions $b_6$ and $c_6$. On the Type IIB side, this moduli space would come from a non-geometric construction, while from the Type IIA perspective it arises, for instance, under compactification on a rigid CY3 ($h^{2,1} = 0$). The metric on this moduli space is given as follows [81]:

$$ds^2_{\mathrm{FH}} = \frac{e^{-\phi} + 2c}{e^{-\phi} + c} d\phi^2 + e^{-\phi}(dc_0^2 + dc_6^2) + \frac{e^{-\phi} + c}{16(e^{-\phi} + 2c)} e^{-2\phi}(db_6 + c_0 dc_6 - c_6 dc_0)^2, \tag{51}$$

where we assumed a square lattice for the middle cohomology of the rigid CY3. The coefficient $c$ here denotes a string one-loop correction [82–84], related to the Euler characteristic of the CY3 as $c = -\chi/(192\pi)$, with $\chi = 2(h^{1,1} - h^{2,1})$. Naively this space looks like a $T^3$ fibration over the real line $\mathbb{R}$ parametrized by $\phi$, where the circles in question are parametrized by the axions. However, notice that there is a non-trivial fibration structure among the axions

themselves—the circle of $b_2$ is fibered over the $T^2$ parametrized by $c_0$ and $c_6$. This gives rise to the following transformations under shifts of the axions

$$(c_0, c_6, b_6) \sim (c_0 + \alpha, c_6 + \beta, b_6 + \gamma + \alpha c_6 - \beta c_0), \tag{52}$$

where we take $\alpha, \beta, \gamma \in \mathbb{Z}$, although the precise quantization condition is model-dependent. This means that the moduli space fiber over $\mathbb{R}$ is not three-torus $T^3$ but is rather a nilmanifold known as the *Heisenberg group* $\mathcal{N}^3 = H^3(\mathbb{R})$. Restricting the axions to an interval corresponds to considering the quotient manifold $\mathcal{N}^3 = H^3(\mathbb{R})/H^3(\mathbb{Z})$ instead. The first Betti number of this space is $b_1(\mathcal{N}^3) = \dim H_1(\mathcal{N}^3, \mathbb{R}) = 2$, as can be seen from the commutation relation $[\partial_{c_0}, \partial_{c_6}] = \partial_{b_6}$. This characterizes an important topological difference between $T^3$ and $\mathcal{N}^3$: Note that $b_1(T^3) = 3$ instead. From a physical point of view, one must therefore be careful in characterizing the spectrum of axionic strings, as this example shows that there are only two axions in a setup that seems naîvely to contain three axionic directions.

## 3.4 Flat moduli spaces

In all of the preceding examples, the moduli spaces were either *hyperbolic* or "approximately" so in a sense that was made precise in [2]. In particular, the authors of [2] conjectured that the asymptotic curvature of moduli spaces in quantum gravity is nonpositive. Though this conjecture was not true in its original stated form [85, 86], it turns out that the spirit of its statement is essentially still preserved in all known examples.[19] This connection was made explicit in [26]: It was proposed that the marked moduli space $\hat{\mathcal{M}}$ necessarily satisfies the so-called *unique geodesic property* – every pair of points can, modulo technical caveats, be connected by a unique geodesic in $\hat{\mathcal{M}}$. The Cartan-Hadamard theorem [89, 90] then links the unique geodesic property to the nonpositivity of moduli space curvature. Moreover, the results of [26] further point towards the fact that "sufficiently far away" from isolated finite-distance singularities in the moduli space interior, the curvature is always nonpositive.

In all the examples discussed previously, not only is the curvature nonpositive "almost everywhere", it is actually *strictly* negative away from finite-distance singularities. However, the strict negativity of curvature is by no means a general feature – there are many examples of moduli spaces with zero curvature, and these examples seem to behave rather differently from their negative-curvature cousins. In this section, we will briefly discuss the properties of flat moduli spaces, elucidating the differences between them and the negative-curvature ones studied earlier.

**Type IIA.** The most basic example of a flat moduli space is Type IIA in 10d. In this case, there is a single modulus – the dilaton $\phi$ – so the moduli space $\mathcal{M} = \mathbb{R}$ is of course flat. The duality group is trivial, rendering $\hat{\mathcal{M}} = \mathbb{R}$ as well. The strong-coupling limit of Type IIA is, as is well-known, a decompactification limit to M-theory in 11 dimensions.

**M-theory on a Klein bottle.** A more nontrivial example of a flat moduli space is furnished by M-theory on a Klein bottle (KB). This theory lies in one disconnected component of the moduli space of rank-1 9d $\mathcal{N} = 1$ supergravity theories, which in another limit reduces to the asymmetric orbifold of Type IIA (AOA) background [91].

The moduli space of M-theory on KB is specified by the geometric moduli of the KB, which can be specified by the radii $r_1, r_2$ of the two independent cycles of the KB. The total moduli space of the effective theory is given by

$$\mathcal{M} = \mathbb{R} \times \mathbb{R}/\mathbb{Z}_2, \tag{53}$$

---

[19]The physical nature of the positive divergence of the curvature has been clarified through the decoupling of field theory sectors in [87, 88].

equipped with the flat metric. Here, the $\mathbb{R}/\mathbb{Z}_2$ factor is parametrized by $\log(r_1 r_2)$, and the quotient by $\mathbb{Z}_2$ reflects the self-T-duality of the AOA theory. Note also that the moduli space can be obtained as the rank-1 Narain double quotient:

$$\mathcal{M} \simeq \mathbb{R} \times \mathrm{SO}(1,1;\mathbb{Z}) \backslash \mathrm{SO}(1,1)/(\mathrm{SO}(1) \times \mathrm{SO}(1)). \tag{54}$$

We note that $\mathrm{SO}(1,1) \simeq \mathbb{R}$ and $\mathrm{SO}(1,1;\mathbb{Z}) \simeq \mathbb{Z}_2$, which acts by negating $\mathrm{SO}(1,1)$. It is in this sense that the moduli space of M-theory on KB should be viewed as a "degenerate" case of the symmetric moduli spaces discussed in 3.1, which in general have constant negative curvature.

The duality group $\Gamma$ is the mapping class group of the KB:

$$\Gamma = \mathbb{Z}_2 \times \mathbb{Z}_2. \tag{55}$$

To see how $\Gamma = \mathbb{Z}_2 \times \mathbb{Z}_2$ acts on the charged spectrum, note that

$$H_1(\mathrm{KB}, \mathbb{Z}) \simeq \mathbb{Z} \times \mathbb{Z}_2. \tag{56}$$

Then, on a given $(a, b) \in \mathbb{Z} \times \mathbb{Z}_2$ there are two separate $\mathbb{Z}_2$ actions:

$$\rho_1 : (a, b) \mapsto (-a, b), \qquad \rho_2 : (a, b) \mapsto (a, b + (a \bmod 2)). \tag{57}$$

The action of $\rho_1$ flips the non-torsion cycle of the KB and therefore leaves the moduli space, which is parametrized solely by the radii of the two 1-cycles, unchanged. Note that although it does not act on the moduli space, this $\mathbb{Z}_2$ does act by flipping the sign of charged objects such as M5 branes wrapped along $a$. Therefore, there is a non-trivially-acting duality vortex associated to this $\mathbb{Z}_2$ duality transformation. However, note that much like the case of the 7-brane with monodromy (19) in the 10d IIB example, a closed loop around this object does not induce an action on the moduli.

The action $\rho_2$, on the other hand, corresponds to a Dehn twist of the KB along the non-torsion cycle, exchanging modes $(1, 1)$ that wrap the torsion cycle with modes $(1, 0)$ that do not. As shown in [91, 92], this is precisely the action on the bosonic spectrum induced by self-T-duality of the AOA background, the strong-coupling limit of which is M-theory on KB. The self-T-duality is reflected in the moduli space of M-theory on KB as the $\mathbb{Z}_2$ quotient in the factor $\mathbb{R}/\mathbb{Z}_2$ in (53). It is also known that M5-branes wrapping a 1-cycle on the compact KB transform faithfully under the $\rho_2$ action, thereby yielding a marked moduli space

$$\hat{\mathcal{M}} = \mathbb{R}^2. \tag{58}$$

As with the hyperbolic examples, we can construct duality vortices associated to $\Gamma$ by constructing a codimension-2 object in the bulk spacetime around which the moduli and the charge lattice are acted on by $\mathbb{Z}_2$ monodromy. This example shows, importantly, that dualities are not restricted to hyperbolic moduli spaces.

However, unlike the hyperbolic moduli spaces, note that the flat examples given here have infinite volume. Although we believe that the flat examples represent in some sense an "edge case" of the more generic hyperbolic behavior, we nevertheless wish to develop a condition that unifies the finite-volume behavior observed in hyperbolic moduli spaces with the infinite-volume flat moduli spaces observed here. It is the purpose of Sec. 5 to develop precisely such a condition. In Sec. 4, we will return to hyperbolic CY moduli spaces to see how the representation-theoretic constraints on duality action observed in 2.3.1 are related to a property of the volume growth of the moduli space. It is a generalization of this property which we will see extends to the flat case as well.

# 4 Compactifiability and semisimple representations

In this section, we will make contact between the features of representation theory of duality groups discussed in Secs. 2.3 and 2.4 and the *geometric* aspects of moduli spaces as studied in Sec. 2.2. In Sec. 4.1, we define a geometric notion of *compactifiability* of moduli spaces, and in Sec. 4.2, we will sketch an explicit proof relating the semisimplicity of duality representations (as observed in Sec. 2.4) to the compactifiability of $\mathcal{M}$. It is for this geometric condition that we will finally be able to propose a general Swampland argument in Sec. 5.

## 4.1 Compactifiability

The notion of *compactifiability* of a space can have different precise meanings depending on the mathematical context. For our purposes, however, a more physical interpretation of compactifiability will be relevant, related to the volume growth properties of moduli spaces. Finiteness of moduli space volume has been observed in many examples [93, 94], and has been proven in general for settings such as Calabi–Yau compactifications [95]. However, there are also obvious counter-examples, such as the moduli space of 10d Type IIA string theory, which is of course $\mathcal{M} = \mathbb{R}$. More generally, it is expected that the flat moduli spaces as in Sec. 3.4 have infinite volume.

We now put forth a compactifiability criterion for *all* moduli spaces in EFTs compatible with quantum gravity. Consider a point $\phi_0 \in \mathcal{M}$ and consider all points within a geodesic distance $D$ from it. This fixes a finite region in moduli space

$$\mathcal{M}_D = \{\phi \in \mathcal{M} \mid d(\phi, \phi_0) \le D\}. \tag{59}$$

For moduli spaces of finite volume we find that $\mathcal{M}_D$ approaches a constant as $D \to \infty$, but even for infinite-volume moduli spaces, we find that it cannot increase *too fast*. Namely, based on argument we will see in Sec. 5 involving the finiteness of the number of ground state upon compactifying the theory to 1d, and string theory examples presented in Sec. 4.3, we propose the following. We say that the moduli space $\mathcal{M}$ is *compactifiable* if the $\mathcal{M}(D)$ as defined above satisfies the following growth condition:

$$\mathrm{Vol}(\mathcal{M}_D) \ll D^{n+\epsilon}, \tag{60}$$

for arbitrary $\epsilon > 0$, where $n = \dim \mathcal{M}$. We call this bound the *compactifiability* condition. This condition forbids exponential growth and growth that is faster than the flat space metric. Indeed, in all of the examples we find, the volume of moduli space grows the fastest when the moduli space is flat, in which case it grows like $D^n$. We will detail all of these examples in section 4.3.

The geometric compactifiability criterion is implied by a mathematical notion of compactifiability discussed in detail in the next section. This definition requires the moduli space $\mathcal{M}$ to be suitably embeddable inside a compact analytic space, inside which the complement of $\mathcal{M}$ is of positive complex codimension. We call this property the *algebraic compactifiability* condition. Algebraic compactifiability is only applicable to complex moduli spaces, but it implies compactifiability in those cases. As we will see, it will be the original, geometric compactifiability condition that will have a natural interpretation from the perspective of the finiteness of 1d vacuum states as argued in Sec. 5.

## 4.2 Semisimplicity from compactifiability: Calabi–Yau case

Having introduced the notion of compactifiability (both geometric and algebraic), we now wish to understand the relationship between the geometric property and the representation

theory of the duality group $\Gamma$ as discussed in Secs. 2.3.1 and 2.4. In the case of Calabi–Yau compactifications of Type II string theory, this relationship can be made explicit. Assuming *algebraic* compactifiability of the complex moduli spaces (as defined in the previous section), it is possible to prove that the associated integral representation of the duality group on charged objects in the vector multiplet of the 4d $\mathcal{N} = 2$ supergravity are semisimple (as defined in Sec. 2.4).

We now briefly sketch this argument relating the semisimplicity of duality representations in 4d $\mathcal{N} = 2$ vector multiplets from the compactifiability of the associated moduli space. To be concrete, we will consider Type IIB on a Calabi-Yau threefold (CY3), but our discussion will actually rely on abstract variations of Hodge structure,[20] depending therefore only on the IR information contained with the supergravity effective action. We will work at a physicist's level of rigor, refraining from providing detailed proofs of our assertions but focusing instead on the logical progression of the argument. For a fully detailed mathematical proof, see Theorem 7.25 in [58] and related results in that same section.

To begin, let us unpack the semisimplicity condition. Recall that a representation $V$ of a group $\Gamma$ is semisimple iff it admits a direct sum decomposition into irreducibles:

$$V = \bigoplus_i V_i, \qquad V_i \text{ irreducible.} \tag{61}$$

In other words, $V$ is semisimple iff for every subrepresentation $V' \subset V$, we can find a complementary representation $V''$ such that $V' \oplus V'' = V$. This is equivalent to the statement that $V' \oplus V/V' \simeq V$ provided that $V/V'$ also forms a subrepresentation. Note that the two representations $V' \oplus V/V'$ and $V$ of $\Gamma$ have the same character, hence the same gauge-invariant observables, it is reasonable that they are physically indistinguishable and hence isomorphic, as discussed in Sec. 2.4.

A direct summand $V'$ would be easy to come by if we were to have a positive-definite inner product $\langle \cdot, \cdot \rangle$ on $V$ respecting the action of $\Gamma$. That is,

$$\langle gv, gv \rangle = \langle v, v \rangle, \qquad g \in G, \quad v \in V. \tag{62}$$

In this case, we could form the orthogonal complement $(V')^\perp$ of $V'$ with respect to $\langle \cdot, \cdot \rangle$, which would provide exactly the required direct summand of $V'$. Unfortunately, such a positive-definite inner product is not immediately apparent for a generic duality representation. Recalling the discussion in Sec. 2.3.1, we note that nothing in the construction of the charge lattice $\Lambda$ implies the existence of a canonical bilinear pairing on $\Lambda$. Thus, it appears that we are somewhat stuck.

However, in the context of 4d $\mathcal{N} = 2$ supergravity, the special geometry furnishes us with exactly the pairing we require. For Type IIB on a CY3 $X$, the charge lattice on which the duality group acts is the middle homology $H^3(X, \mathbb{Z})$. Tensoring with $\mathbb{C}$, we obtain a complex representation for the action of the duality group on $H^3(X, \mathbb{C})$. Given forms $\alpha, \beta \in H^3(X, \mathbb{C})$, we now define

$$(\alpha, \beta) = \int_X \alpha \wedge \beta. \tag{63}$$

This pairing respects the action of the monodromy $\Gamma$ on $\alpha, \beta$, i.e. $(g\alpha, g\beta) = (\alpha, \beta)$ for all $g \in \Gamma$. However, it is *not* a positive-definite inner product, but rather skew-symmetric $(\beta, \alpha) = -(\alpha, \beta)$. However, we can construct an associated positive-definite inner product by including the action of the Hodge star automorphism $C : H^3(X, \mathbb{C}) \to H^3(X, \mathbb{C})$, acting by

$$\omega^{p,q} \in H^{p,q} : \qquad C\omega^{p,q} = i^{p-q}\omega^{p,q}. \tag{64}$$

---

[20]In fact, the theorem of [58] about semisimple monodromy groups does not even require the Hodge structure to be of Calabi–Yau type, as it can have any Hodge numbers.

We then define our inner product via

$$\langle \alpha, \beta \rangle = (\alpha, C\beta), \tag{65}$$

which is positive-definite, i.e. $\langle \alpha, \overline{\alpha} \rangle > 0$ for all $0 \neq \alpha \in H^3(X, \mathbb{C})$. This can be seen directly from the Hodge decomposition, as the polarization of the subspaces amounts to $i^{p-q}(\omega^{p,q}, \overline{\omega}^{p,q}) > 0$ for all $0 \neq \omega^{p,q} \in H^{p,q}$.

This is *almost* sufficient to establish semisimplicity of monodromy representations. To see this, note that subrepresentations $V \subset H^3(X, \mathbb{C})$ of $\Gamma$ are in one-to-one correspondence with $\Gamma$-invariant subspaces $V \subset H^3(X, \mathbb{C})$. Given such a $\Gamma$-invariant subspace $V$, we could form its orthogonal complement $V^\perp$ with respect to $\langle \cdot, \cdot \rangle$. But the formation of this orthogonal complement would only respect the action of $\Gamma$ if the following were true:

$$V \text{ invariant under } \Gamma \quad \Longleftrightarrow \quad CV \text{ invariant under } \Gamma. \tag{66}$$

That is, we require $\Gamma$-invariant subspaces to be stable under the action of $C$ as a sufficient condition for the semisimplicity of the monodromy group.

We have therefore reduced the semisimplicity of $H^3(X, \mathbb{C})$ to a condition on the action of the Hodge star operator $C$ with respect to the monodromy group $\Gamma$. Let us take a moment to interpret this condition. Recall from (64) that $C$ acts on the $(p, q)$-subspaces $H^{p,q}$ of $H^3(X, \mathbb{C})$ by multiplication by $i^{p-q}$. Requiring $\Gamma$-invariant subspaces to be $C$-stable thus effectively formalizes the intuitive notion that $\Gamma$ acts "democratically" with respect to this Hodge decomposition. More specifically, even if $C$ acts non-trivially on a given $\Gamma$-invariant vector space $V$—for instance, if $V$ is spanned by vectors of indefinite $(p, q)$-type[21]—the resulting vector space $CV$ remains $\Gamma$-invariant. In the remainder of this subsection we will explain how the mathematical notion of compactifiability precisely implies this democratic action of the monodromy group.

Given a subrepresentation $V \subseteq H^3(X, \mathbb{C})$, we can consider the corresponding $\Gamma$-invariant bundle $E \to \mathcal{M}$ over the moduli space. We wish to show that $CE$ is also $\Gamma$-invariant. This can be more generally understood as follows. Note that $\mathcal{M}$ is equipped with a vector bundle, the fiber of which is $H^3(X, \mathbb{C})$ over each point corresponding to $X$. This bundle is known as the *Hodge bundle* over $\mathcal{M}$. In general, the fundamental group $\pi_1(\mathcal{M})$, which we have seen can be identified with the duality group $\Gamma$, acts on the fibers of this vector bundle. Moreover, a flat vector bundle is then associated isomorphically to a representation of $\Gamma$. Thus, we would like to show that the $\Gamma$-action on flat sections of the Hodge bundle are in some sense "democratic" with respect to the $(p, q)$-decomposition. Modulo technical considerations,[22] it would therefore suffice to show that flat subbundles of the Hodge bundle over $\mathcal{M}$ necessarily decompose into flat $(p, q)$-components.

To summarize, we have turned a statement about the semisimplicity of representations of $\Gamma$ on the charges $H^3(X, \mathbb{C})$ into a statement about flat subbundles of the Hodge bundle over $\mathcal{M}$. The intuition that $\Gamma$ acts democratically with respect to the $(p, q)$-components of $H^3(X, \mathbb{C})$ plays a crucial role in this connection. However, to make further progress, we need to be able to effectively study the $(p, q)$-flat bundles over $\mathcal{M}$. It is precisely here that the *geometry* of $\mathcal{M}$ becomes relevant.

Suppose we have a flat section $e$ of the Hodge bundle. The vector $e$ admits a direct sum decomposition into its $(p, q)$-components:

$$e = \sum_{p+q=3} e_{p,q}. \tag{67}$$

---

[21]By indefinite $(p, q)$-type we mean that a vector $v \in V$ must be spanned by multiple $(p, q)$-components, i.e. $v = \sum_{p,q} v_{p,q}$ where multiple $v_{p,q} \in H^{p,q}$ are non-vanishing.

[22]The difficulty in this part of the proof lies in the fact that we are working with vector subspaces that are invariant under $\Gamma$, which does not necessarily mean that their basis vectors themselves are invariant: they may pick up non-trivial phases by acting with elements of $\Gamma$, or even be mapped into one another. We refer to [58] for a careful treatment of these aspects.

Now, we can again make use of our positive-definite inner product over $H^3(X, \mathbb{C})$, with which we may define the following functions $\phi_{p,q} : \mathcal{M} \to \mathbb{R}_{\geq 0}$:

$$\phi_{p,q} = \langle e_{p,q}, e_{p,q} \rangle. \tag{68}$$

That is, $\phi$ measures the square lengths of the $(p, q)$-components of $e$. In fact, we can say more about the functions $\phi_{p,q}$: If they are nonconstant, they must satisfy certain positivity and growth conditions such that they each define global potentials over $\mathcal{M}$. In the mathematics literature, the $\phi_{p,q}$ are called *plurisubharmonic functions*.

Here is where the geometry of $\mathcal{M}$ appears. We introduced a definition of compactifiability based on regular volume growth of $\mathcal{M}$ in Sec. 4.1. For a CY vector multiplet moduli space $\mathcal{M}$, which are special Kähler manifolds, we can further impose *algebraic compactifiability*; that is, we demand that the asymptotic boundary of $\mathcal{M}$ have positive complex codimension.[23] This condition implies our weaker notion of compactifiability based on the asymptotic volume growth of $\mathcal{M}$. Intuitively, if the boundary has positive complex codimension, then it has real codimension at least two, meaning that the boundary must "pinch" off in such a way that the volume growth is regulated. Near such pinched boundary, we may show that $\phi_{p,q}$ must have certain standard asymptotical behaviour independent of the finite part of the moduli space. In particular, we may show that they are bounded from above. Algebraic compactifiability guarantees that all the boundaries are pinched, hence $\phi_{p,q}$ are bounded from above near all the boundaries, and hence bounded above on the entire $\mathcal{M}$.

On the other hand, if $\mathcal{M}$ is algebraically compactifiable, then it is known that any global potential over $\mathcal{M}$ must be *unbounded from above*. Indeed, this follows from a basic property of plurisubharmonic functions named the maximum principle [96]. Thus, we appear to have tension between the boundedness of the square lengths of $e_{p,q}$ over $\mathcal{M}$ and the global geometry of $\mathcal{M}$. The only way out is that $\phi_{p,q}$ be *constant*, in which case the $e_{p,q}$ are certainly flat. This, as we have discussed, implies our representation-theoretic characterization of $\Gamma$, as desired.

To summarize once more, we have transformed a representation-theoretic condition on $\Gamma = \pi_1^{\text{orb}}(\mathcal{M})$ into a geometric condition on the flat vector bundles over $\mathcal{M}$. We then used the algebraically compactifiable geometry of the base $\mathcal{M}$ to constrain its flat bundles to have flat $(p, q)$-decompositions, thereby deducing the semisimplicity of representations of $\Gamma$.

We conclude this section with a small remark. The above discussion relied on the concrete top-down construction of $\mathcal{M}$ as the complex structure moduli space of a CY3. However, with sufficient care, it can be formulated abstractly entirely in terms of variations of Hodge structure, independently of top-down descriptions. From this picture, with the algebraic compactifiability of $\mathcal{M}$ as an input, it is possible to deduce the semisimplicity of $\Gamma$-representations. Thus, our argument furnishes a relationship between semisimplicity and compactifiability purely at the level of the IR supergravity theory for 4d $\mathcal{N} = 2$ theories. In particular, it does not rely on the geometric structure provided by the Calabi–Yau manifold, so in principle it extends also to non-geometric constructions such as string orbifolds, provided of course that these moduli spaces are also compactifiable (in the algebraic sense). In the next section, we will illustrate a simple example where the failure of $\mathcal{M}$ to be compactifiable is seen to be related to the non-semisimplicity of duality representations.

## 4.3 Examples

Having discussed compactifiability and the semisimplicity of duality groups in the context of Calabi–Yau moduli spaces, let us now turn to some explicit examples. We want to showcase both how familiar moduli spaces in string theory are compactifiable (in the geometric sense) and connect this compactifiability property directly with the semisimplicity of duality representations.

---

[23]To be precise, we demand that $\mathcal{M}$ be Zariski-open in a compact analytic space.

### 4.3.1 Upper-half plane

The stage of our first set of examples is given by the upper-half plane, on which we consider the action of various duality groups $\Gamma \subseteq SL(2, \mathbb{Z})$ and the corresponding quotients. The metric on $\mathbb{H}$ is the usual hyperbolic metric

$$ds^2 = \frac{d\tau_1^2 + d\tau_2^2}{\tau_2^2}. \tag{69}$$

The geodesic distance between two points $\tau = \tau_1 + i\tau_2$ and $\rho = \rho_1 + i\rho_2$ is given by

$$d(\tau, \rho) = 2\mathrm{ArcSinh}\left[ \frac{\sqrt{(\tau_1 - \rho_1)^2 + (\tau_2 - \rho_2)^2}}{2\sqrt{\tau_2 \rho_2}} \right]. \tag{70}$$

In all examples we will consider $\rho = x + iy = i$ as starting point and move away from this point by a geodesic distance $D$. We then wish to compute the volume of the set of all points within this geodesic distance, and determine the scaling of this volume in $D$.

**Trivial duality group.** To warm up, we first consider the case where the duality group is trivial, i.e. $\Gamma = \{1\}$, so we consider the whole upper-half plane $\mathcal{M} = \mathbb{H}$. As mentioned above, our reference point is taken to be $\rho = i$, the would-be self-duality point of the usual S-duality, which is not present now. Then the effective moduli space $\mathcal{M}_D$ is given by

$$\mathcal{M}_D = \{\tau \in \mathbb{H} \mid \tau_1^2 + (\tau_2 - \cosh D)^2 = \sinh^2 D\}. \tag{71}$$

In other words, $\mathcal{M}_D$ is a disk in $\mathbb{H}$, which is centered at $\tau = i\cosh D$ with radius $\sinh D$. In order to compute its bulk volume, we have to use the metric (69), which yields

$$\mathrm{Vol}(\mathcal{M}_D) = 2\pi(\cosh D - 1). \tag{72}$$

Thus we find that the volume increases exponentially with the geodesic distance $\Delta$, violating our compactifiability criterion (60).

**Upper-triangular duality group.** As our next example we consider the upper-triangular duality group generated by the transformation $\tau \mapsto \tau + 1$; that is, $\Gamma = \mathbb{Z} = \langle T \rangle$. We expect that the associated quotient will not be compactifiable due to the infinite-distance asymptotic boundary at the real line $\mathrm{Im}(\tau) = 0$. Before we compute the volume of this moduli space, however, let us check how exactly the proof [58] that we reviewed in Sec. 4.2 breaks down. The invariant subspace of this duality group is given by

$$L = \mathrm{span}_{\mathbb{C}}\{(1, 0)\}. \tag{73}$$

Let us now express $L$ in terms of the period vector $\mathbf{\Pi} = (1, \tau)$ of the $(1, 0)$-form, which yields

$$L = \mathrm{span}_{\mathbb{C}}\left\{ \frac{i}{2\tau_2}(\overline{\tau}\mathbf{\Pi} - \tau\overline{\mathbf{\Pi}}) \right\}. \tag{74}$$

One way to see the failure of semisimplicity as argued for in Sec. 4.2 is to consider the action of the Hodge star operator $C$ on $L$. Acting with the Hodge star operator $C$ amounts to $C\mathbf{\Pi} = i\mathbf{\Pi}$ and $C\overline{\mathbf{\Pi}} = -i\overline{\mathbf{\Pi}}$, so we find that

$$CL = \mathrm{span}_{\mathbb{C}}\left\{ \frac{1}{2\tau_2}(\overline{\tau}\mathbf{\Pi} + \tau\overline{\mathbf{\Pi}}) \right\} = \mathrm{span}_{\mathbb{C}}\left\{ \frac{1}{2\tau_2}(\tau_1, 2|\tau|^2) \right\}. \tag{75}$$

Clearly this is no longer an invariant subspace under the monodromy group, so it violates (66) of the argument of the previous subsection, which states that monodromy-invariant subspaces should be stable under the Hodge star.

As a result, one need not expect that the potential obtained in Eq. 67 be bounded over the moduli space. We also show this explicitly as follows. Consider the potential of Eq. (67) related to $L$, which we recall is defined as the norm of the $(1,0)$-form component spanning $L$. In this case, we find that

$$\varphi = i(L^{1,0}, \overline{L^{1,0}}) = \frac{i|\tau|^2}{4\tau_2^2}(\mathbf{\Pi}, \overline{\mathbf{\Pi}}) = \frac{|\tau|^2}{2\tau_2} \,. \tag{76}$$

Compactifiability of the moduli space requires that the potential $\varphi$ be bounded everywhere in moduli space; this is clearly not the case here, as the right hand side of Eq. (76) diverges as $\tau_2 \to \infty$.

Having illustrated the mathematical proof of how compactifiability implies semisimplicity, let us now compute the volume of this moduli space directly; we wish to see how it grows for large distances $D \gg 1$. The fundamental domain corresponding to our duality group $\Gamma = \langle T \rangle$ is as follows:

$$\mathcal{M}_{\text{strip}} = \mathbb{H}/\langle T \rangle = \{\tau \in \mathbb{H} | -1/2 \leq \tau_1 < 1/2\} \,. \tag{77}$$

We then wish to compute the volume of the region inside $\mathcal{M}_{\text{strip}}$ which lies within a geodesic distance $D$ of $\tau = i$. For sufficiently large $D$ this region is given by

$$\mathcal{M}_D = \left\{\tau \in \mathbb{H} \middle| -\frac{1}{2} \leq \tau_1 \leq \frac{1}{2}, \ e^{-D} \leq \tau_2 \leq e^D\right\} \,. \tag{78}$$

The volume of this moduli space is then

$$\text{Vol}(\mathcal{M}_D) = e^D + \mathcal{O}(e^{-D}) \,. \tag{79}$$

This exponential growth in $D$ violates our compactifiability criterion, therefore ruling out $\Gamma = \langle T \rangle$ as a possible duality group. In other words, we learn that we need to include additional dualities, such as S-duality, in order to be able to properly compactify our moduli space.

**Full SL(2, $\mathbb{Z}$) duality group.** After these examples of non-compactifiable moduli spaces, let us now consider the usual fundamental domain of SL(2, $\mathbb{Z}$). The total volume of this moduli space is well-known to be finite and given by

$$\text{Vol}(\mathbb{H}/\text{SL}(2, \mathbb{Z})) = \frac{\pi}{3} \,. \tag{80}$$

From this we can already conclude that the moduli space must be compactifiable, but for completeness let us nevertheless determine the scaling of the volume with the geodesic distance $D$. This amounts to subtracting an amount $e^{-D}$ for the region between $\tau_2 = e^D$ and $\tau_2 = \infty$, which yields

$$\text{Vol}(\mathcal{M}_D) = \frac{\pi}{3} - e^{-D} \,. \tag{81}$$

Thus the moduli space is compactifiable, and the volume approaches its finite value exponentially fast in the geodesic distance.

**Fuchsian groups of the first kind.** Having seen that $\Gamma = \text{SL}(2, \mathbb{Z})$ results in a fundamental domain $\mathbb{H}/\Gamma$ with finite volume, one may ask in general what subgroups $\Gamma \subseteq \text{SL}(2, \mathbb{R})$ have this property. Assuming that $\Gamma$ is finitely generated, this finite-volume property has been shown to be equivalent to the property that $\Gamma$ be a Fuchsian group of the first kind (see e.g. [97] for details). Such a group is defined by stipulating that its limit set $\Lambda(\Gamma)$ – the set of limit points of $\Gamma z$ for $z \in \mathbb{H}$ – be equal to

$$\Lambda(\Gamma) = \mathbb{R} \cup \{\infty\}. \tag{82}$$

In other words, if we consider the fundamental domain $\mathbb{H}/\Gamma$, then it can only extend downwards through cusps that reach $\mathbb{R}$ at a single point. This also fits with our observation from the volume of the strip (77), since a fundamental domain that extends to $\mathbb{R}$ with a finite width always has infinite volume.

### 4.3.2 4d $\mathcal{N} = 2$ moduli spaces

We now turn to the moduli spaces of Calabi–Yau compactifications of type II string theories. Although we have already discussed the vector multiplet geometry and charged spectrum in great detail in section 4.1, we will find it useful to consider an explicit metric and study the potential corrections to this metric.

**Vector multiplet.** In Type IIA (resp. Type IIB) string theory Calabi–Yau compactifications, the vector multiplet moduli space can be shown to be hyperbolic in the large volume/LV (resp. large complex structure/LCS) limit. For instance, consider compactifying Type IIA on the quintic (resp. Type IIB on the mirror quintic), there is a single complex modulus, and the metric on the moduli space reduces to:

$$ds^2 \sim d\phi^2 + e^{-2\phi} da^2, \tag{83}$$

where the modulus $z = a + i e^{\phi}$ is a Kähler (resp. complex structure) modulus. This metric is only valid in the LV/LCS limit, where $\phi$ is large. The axion is periodic, so in the limit $\phi \to \infty$, we see that the metric appears to yield a compactifiable geometry, since the size of the axionic circle decreases exponentially as $e^{-2\phi}$. However, for the *global* moduli space geometry to be compactifiability, we have to also worry about the opposite limit, as one goes into the bulk, away from the LV (resp. LCS) limit.

From Sec. 4.1, we know that the volume of vector multiplet moduli space is finite, it must therefore be the case that various corrections to the metric away from the LV (resp. LCS) limit are responsible for this. As one goes into the bulk, the metric receives corrections that are purely classical in the Type IIB picture but correspond to $\alpha'$ corrections and non-perturbative worldsheet instanton corrections in the mirror-dual Type IIA picture. In particular, the resummation of these worldsheet instanton corrections in the moduli space interior results in a finite distance conifold singularity and Landau-Ginzburg orbifold singularity instead. Note that this is a classic instance of a situation where instantons are ultimately responsible for compactifying the moduli space geometry as we described in Sec. 2.3.2. We will find a similar phenomenon in the case of the hypermultiplet moduli space of 4d $\mathcal{N} = 2$ compactifications which is detailed below.

In the case of the (mirror) quintic moduli space, the resummation of worldsheet instantons modify the metric such that only finite distance singularities remain in the moduli space interior, this is not the case in general. Indeed, in other examples, further infinite-distance singularities may also appear. In fact, among the fourteen Calabi–Yau threefolds with $\mathcal{M}_{cs}$ listed in Table 2, there are four cases where instead of a Landau-Ginzburg orbifold singularity, one finds an infinite distance singularity – either a K-point or another LCS (maximal unipotent monodromy) point. Around these singularities we can bring the metric again into the

negative-curvature hyperbolic form (83), similar to the LV/LCS regime. This results in a finite contribution to the volume when taking the shrinking axionic radius into account. In fact, this hyperbolic behavior of the metric close to boundary loci is a general feature of complex structure moduli spaces of Calabi–Yau manifolds [2]. The underlying structure has been extensively studied through asymptotic Hodge theory [53], particularly in the context of the distance conjecture.

A general proof of the finiteness of the moduli space volume of Calabi–Yau manifolds was given in [98]; in fact, in [95] it was shown that these volumes are actually rational numbers (up to factors of $\pi$). These proofs crucially uses that the moduli space of a Calabi–Yau manifold is quasi-projective [99]. For our purposes, this means that singularities in moduli space can be described as normal-crossing divisors, which can locally be described as $z^1 = \ldots = z^n = 0$. The phase of these coordinates $z^i$ may be identified with the axion, while the absolute value corresponds to the saxion. Using these physical coordinates, it then follows from the nilpotent orbit theorem of [58] that the geometry becomes asymptotically hyperbolic, resulting in a finite contribution to the volume coming from each boundary component.

To see that the volumes are rational, let us consider the 14 hypergeometric examples listed in table 2. Firstly, it is helpful to note that the Kähler potential is defined globally over these moduli spaces and not patch-wise. This was argued on general grounds on general grounds for 4d $\mathcal{N} = 2$ supergravity [100] by demanding anomalies to be well defined, and verified for toroidal orbifolds in [101]. In the 14 cases with $\mathcal{M}_{cs} = \mathbb{P}^1 - \{0, 1, \infty\}$ it follows from the fact that we do not need to shift the Kähler potential as we transition between the patches around $z = 0, 1, \infty$, we just match series expansions of the periods between neighboring regions. This allows us to use Stokes theorem to compute the volume of the moduli space, see for instance [102] for $\mathbb{H}/SL(2, \mathbb{Z})$. We rewrite the volume integral into contour integrals around the punctures as[24]

$$\mathrm{Vol}(\mathcal{M}_{cs}) = \tfrac{1}{2} \int_{\mathcal{M}_{cs}} \partial \overline{\partial} K = \lim_{\epsilon \to 0} \sum_{z_s = 0, 1, \infty} \tfrac{1}{2} \int_{S_\epsilon^1(z_s)} \partial K \,, \tag{84}$$

where we take the limit of vanishing radius $\epsilon \to 0$. We need to be careful here to make sure that $\overline{\partial} K$ is globally defined and does not transform under monodromies around the singularities. This amounts to fixing a gauge for the Kähler transformations $K \to K - \log |f(z)|^2$, since under holomorphic rescalings of the $(3,0)$-form $\Omega \to f(z)\Omega$ the integrand in (84) picks up a piece $\partial f / f$. In particular, we should **not** divide by the fundamental period $\Pi^0(z)$ as is done customarily, as this would introduce a non-invariant piece $\partial \Pi^0(z)/\Pi^0(z)$ at $z = \infty$; instead, we stick with the periods obtained directly from the Picard-Fuchs differential equation. In that Kähler frame the Kähler potential behaves asymptotically at the large complex structure and conifold point as

$$K_{\mathrm{LCS}} = -\log\left[ \frac{\kappa}{(2\pi i)^3} \log |z_{\mathrm{LCS}}|^3 + \mathcal{O}(1) \right], \qquad K_{\mathrm{C}} = -\log\left[ a + b|z_c|^2 \log |z_c|^2 + \mathcal{O}\left(|z_c|^2\right) \right]. \tag{85}$$

where we introduced the local LCS and conifold coordinates $z_{\mathrm{LCS}} = 1/z$ and $z_c = z - 1$. The numbers $\kappa \in \mathbb{Z}$ and $a, b \in \mathbb{C}$ depend on the example under consideration [61, 104, 105], see for example [106] for a recent review and detailed notebooks. For the contour integrals we find that the integrands $\partial K_{\mathrm{LCS}}$ and $\partial K_{\mathrm{C}}$ fall off too quickly, so only the integral around $z = 0$ contributes. There we can extract a piece $-\log |z|^{2a_1}$ as leading contribution from the Kähler potential, and the remainder does not contribute in a way similar to the LCS and conifold point. This leaves us with the volume

$$\mathrm{Vol}(\mathcal{M}_{cs}) = \lim_{\epsilon \to 0} \int_{S_\epsilon^1(0)} \frac{a_1}{z} dz = 2\pi a_1 \,. \tag{86}$$

---

[24]See [103] for a similar calculation of an index for flux vacua.

The volumes of the moduli space are thus determined directly by the leading exponent $a_1$ of the periods. We have included these values in table 2. It is remarkable that the volume only depends on this boundary information, and in particular not even the leading coefficients in the Kähler potentials are needed.

**Hypermultiplet.** Let us now address the case of instantons and compactifiability for the hypermultiplet sector. For simplicity, we consider again the case a CY3 with two hypermultiplets, deferring a more detailed discussion to future work. Recall from section 3.3 that this moduli space is parametrized by four R-R axions $c_0, c_2, c_4, c_6$, two NS-NS axions $b_2, b_6$, and two more scalars $\phi$ and $r$ corresponding to the string coupling and the 2-cycle volume of the Calabi-Yau. The metric on this moduli space is given by [81]

$$ds^2_{\text{HM}} = d\phi^2 + 4ds^2_{\mathcal{M}_K} + e^{-\phi} ds^2_{T^4} + \frac{1}{16}e^{-2\phi}(db_6 + c_0 dc_6 + c_2 dc_4 - c_6 dc_0 - c_4 dc_2)^2, \quad (87)$$

where $ds^2_{\mathcal{M}_K}$ denotes the line element on the complexified Kähler moduli space of the CY3, and $ds^2_{T^4}$ denotes the line element of the four-torus parametrized by $c_0, c_2, c_4, c_6$. Note that we have shown only the leading-order metric, already having suppressed the one-loop correction governed by the Euler number of the CY3.

Let us now verify whether this moduli space is compactifiable along the weak-coupling and large-volume limits $\phi \to \infty$ and $r \to \infty$. In the large-volume limit $r \to \infty$ we find that the metric only depends on $r$ through the metric $ds^2_{\mathcal{M}_K}$ on the Kähler moduli space. We already know from the above discussion of the vector multiplet geometry that this space has finite volume; thus, we find that the hypermultiplet metric $ds^2_{\text{HM}}$ must also be compactifiable along this limit. Along the other limit, the weak-coupling limit $\phi \to \infty$, we find that the volume form scales as $e^{-3\phi}$, so it is also compactifiable. Thus, we see that the hypermultiplet moduli space is compactifiable in the usual perturbative limits.

When we move away from this perturbative regime, in particular $\phi \to -\infty$, the naive perturbative metric (87) yields a moduli space with an exponentially growing volume. Thus compactifiability requires that either dualities or corrections exclude this part of the moduli space and/or alter the volume growth. To this end, let us first consider how the string one-loop correction affects the metric (51) of the fundamental hypermultiplet moduli space. We find that this correction obstructs us from taking the strong-coupling limit $e^\phi = \infty$, as we instead run into a metric singularity at $e^\phi = -1/(2c)$.[25] In other words, this region of exponentially large volume is excluded, and instead we appear to have another end of moduli space with finite volume. Note, however, that there are further corrections to the moduli space metric such as D-instantons that become large in this regime, and are therefore crucial in determining the ultimate fate of this singularity.

Further evidence for the compactifiability of the hypermultiplet moduli space in such regimes is provided by the study of certain strong-coupling/large volume limits in [49]. The authors of [49] considered trajectories along which the volume $r$ and 10d dilaton $\phi$ are double-scaled in such a way that the (classical) 4d dilaton $e^{-\phi} r^{3/2}$ is kept fixed. It was found that D-instantons correct the hypermultiplet metric such that these limits are brought to finite distance. In particular, from the finiteness of these paths, we find also that the volume of the moduli space also cannot diverge locally along these strong-coupling limits into the non-perturbative interior of the moduli space.

The analysis of [49] was further extended in [107] by considering different scalings of the Kähler volumes of the Calabi–Yau threefolds while still holding the classical 4d dilaton fixed. From this analysis, it was realized a class of trajectories in fact remain at infinite distance despite the D-instanton corrections. The tower of light states, as required by the Distance

---

[25]Note that the right-hand side is positive, as $c < 0$ for a rigid CY3 ($h^{2,1} = 0$).

Conjecture, is furnished by a D3-brane wrapping a 2-cycle, dual to a tensionless fundamental Type IIB string. It would be interesting to check whether the moduli space volume also remains finite along these infinite distance limits.

### 4.3.3 M-theory and type IIA

Finally, we consider M-theory and Type IIA as straightforward examples. These examples have simple duality groups and flat moduli spaces with possibly infinite volume. Nevertheless, as we will see, the compactifiability condition is still satisfied in all the examples we check.

**M-theory.** M-theory has no moduli, so the moduli space $\mathcal{M}$ can be thought of as a set with a single point (or a set of points): $\mathcal{M} = \{*\}$. The volume of this zero-dimensional space is given by the number of elements in this set, so $\mathrm{Vol}(\mathcal{M}) = 1$. This fits with our compactifiability criterion in a trivial way, since the volume is finite.

**Type IIA.** In Type IIA, we have a one-dimensional moduli space parametrized by the dilaton $\phi$, yielding the geometry $\mathbb{R}$ with the usual metric. The volume of a geodesic ball of radius $D$ in this moduli space thus grows as

$$\mathrm{Vol}(\mathcal{M}_D) = 2D. \tag{88}$$

This moduli space is also compactifiable, since its volume scales linearly with geodesic distance.

**M-theory on a Klein bottle.** As discussed in section 3.4, the moduli space of M-theory on the Klein bottle is the two-dimensional moduli space $\mathcal{M} = \mathbb{R} \times \mathbb{R}/\mathbb{Z}_2$, with a flat metric. In this case it is clear that the boundary of moduli space is infinitely large, but let us nevertheless consider how fast this boundary grows as one goes to an infinite distance limit. Following the notation of section 3.4 and taking $D \sim \sqrt{r_1^2 + r_2^2}$ as a sort of radial coordinate on moduli space, it is clear that the volume of a geodesic ball goes as follows:

$$\mathrm{Vol}(\mathcal{M}_D) \sim D^2. \tag{89}$$

Since $n = \dim \mathcal{M} = 2$ in this example, we see that the compactifiability bound is satisfied.

## 5 Compactifiability and finiteness

In this section, we put all the pieces together for the final denouement. Motivated by the semisimplicity of duality representations (Sec. 2.4) and its relation to the compactifiability of moduli spaces (Sec. 4), we now argue for the compactifiability condition from the bottom-up. In particular, we will use the *principle of finiteness* to argue for compactifiability. Using the principle that there must be a finite number of massless states upon compactifying all spatial dimensions, we argue that if the volume of moduli space is infinite, then the volume growth of the moduli space with geodesic distance can be no faster than Euclidean space.

The finiteness of string vacua based on Calabi–Yau manifolds was conjectured long ago by Yau [108] and the idea that the number of quantum gravity vacua should be finite was independently proposed in [93, 109] and subsequently refined in [20, 110]. Significant progress has recently been made in understanding these finiteness aspects of the landscape, including 6d supergravity theories through physical arguments [111–119] and the landscape of F-theory flux vacua using methods from tameness [78, 79, 120–122].

Suppose we are given a consistent EFT coupled to gravity with an exact moduli space $\mathcal{M}$. Let us fix a point $\phi_0 \in \mathcal{M}$ and consider, as in Sec. 4.1, the geodesic ball of radius $D$; that is, consider the region within geodesic distance $D$ from $\phi_0$:

$$\mathcal{M}_D := \{\phi \in \mathcal{M} \mid d(\phi, \phi_0) \leq D\}, \tag{90}$$

where we suppress the dependence on the initial point $\phi_0$ (which is immaterial in the limit $D \to \infty$).

Recall also that the species scale $\Lambda$ can be regarded as a function of the moduli [5]. We may therefore alternatively define the moduli space volume regulated by the species scale $\Lambda$:

$$\mathcal{M}_\Lambda := \{\phi \in \mathcal{M} \mid \Lambda(\phi) \geq \Lambda\}. \tag{91}$$

Note that the two notions of "regulated" moduli space $\mathcal{M}_\Lambda$ and $\mathcal{M}_D$ are related by the species scale form of the Distance Conjecture, which posits that

$$\Lambda \simeq e^{-\beta D}, \tag{92}$$

as $D \to \infty$ for some constant $\beta \sim \mathcal{O}(1)$.

The volume of the regulated moduli spaces $\mathcal{M}_\Lambda$ and $\mathcal{M}_D$ can then be regarded as functions of $\Lambda$ and $D$, which we denote as follows:

$$V(\Lambda) = \text{Vol}(\mathcal{M}_\Lambda), \qquad V(D) = \text{Vol}(\mathcal{M}_D). \tag{93}$$

Letting $n = \dim \mathcal{M}$, our central claim is that $\mathcal{M}$ is compactifiable, which we recall to mean that

$$\boxed{\lim_{D \to \infty} V(D) \ll D^{n+\epsilon},} \tag{94}$$

for arbitrarily small $\epsilon > 0$. In other words, the volume of a geodesic ball in $\mathcal{M}$ must grow no faster than that of a ball in Euclidean space.

To connect this volume growth property with finiteness of amplitudes in quantum gravity, recall first that in [123, 124] it was argued that the moduli space of 0-branes in a consistent quantum gravity theory must be compact in order to be consistent with the finiteness of the number of light states. The bulk moduli fields of supergravity theories, on the other hand, cannot be viewed analogously to the moduli of a 0-brane, because they can be frozen at a given supergravity vacuum in $d > 2$ spacetime dimensions. However if we compactify down to $d \leq 2$, the vevs of these scalars fields cannot be set to a fixed value, putting them on a similar footing to the 0-brane moduli of [123, 124]. In particular if we compactify all spatial dimensions (for instance, on a torus), the low-energy dynamics of the resulting 1d theory will include a sigma model with target space given by the moduli space of scalars. This was used in [20]—compactifying down to 1d, the authors showed that $V(\Lambda) < \infty$ *at a fixed* $\Lambda$. Here, we wish to refine this argument in order to deduce to our desired compactifiability bound as $\Lambda \to 0$.

Our task then is to understand how the finiteness of the number of ground states of the sigma model with target space[26] $\mathcal{M}$ places a restriction on the volume growth of $\mathcal{M}_\Lambda$ as one approaches its asymptotic boundary $\Lambda \to 0$. Indeed, when we take $\Lambda \to 0$, the ground states of the sigma model are the only states in the EFT description that we may expect to make

---

[26]To be precise, the moduli space of scalars $\mathcal{M}$ gets enlarged as we compactify all spatial dimensions, embedding within some larger space $\widetilde{\mathcal{M}}$. Our statements, while technically applying to the volume growth of $\widetilde{\mathcal{M}}$, will also carry over to $\mathcal{M}$. This is due to the fact that the embedding $\mathcal{M} \hookrightarrow \widetilde{\mathcal{M}}$ is isometric, at least asymptotically in the additional moduli directions. More specifically, there can be instanton corrections to the $\mathcal{M}$ metric upon a torus compactification due to Euclidean branes wrapping the torus, but these are insignificant in the large-volume limit of the compact torus. Thus, if $\mathcal{M}$ is non-compactifiable, then so is $\widetilde{\mathcal{M}}$.

good sense in a duality invariant setup. In particular, higher energy states will be sensitive to the tower of light states present in the $\Lambda \to 0$ limit. Provided the sigma model has at least two real supercharges (which would be the case if the higher dimensional theory has any supersymmetry for $d > 2$), then its space of ground states is known from supersymmetric quantum mechanics [125] to be the space of harmonic $p$-forms on $\mathcal{M}$ for all $0 \le p \le \dim \mathcal{M}$ (up to finite multiplicity depending on the amount of supersymmetry). For example, if a $p$-form is written in local coordinates as $f(X_i)dX_1 \wedge ...dX_p$, then it corresponds to a state $f(X_i)\psi_0^{(1)}...\psi_0^{(p)}|0\rangle$ where $\psi_0^{(i)}$ are the zero-modes for the superpartners to the $X_i$ fields. See [125] and the review in Chapter 10 of [126] for more details.

For such harmonic $p$-forms to correspond to physical vacuum states, these states must be normalizable. For a state $|\alpha\rangle$ corresponding to a $p$-form $\alpha_p \in \Omega^p(\mathcal{M})$, this means that

$$\langle \alpha | \alpha \rangle = \int_{\mathcal{M}} \alpha_p \wedge *\alpha_p < \infty \,. \tag{95}$$

Such forms are known in the math literature as $L^2$-normalizable, and we denote the set of such $p$-forms on $\mathcal{M}$ with metric $g_{\mathcal{M}}$ by $\mathcal{H}_{L^2}^p(\mathcal{M}, g_{\mathcal{M}})$. Therefore, consistency with the finiteness principle implies that

$$\dim \mathcal{H}_{L^2}^p(\mathcal{M}, g_{\mathcal{M}}) < \infty \,, \quad \text{for all } p \,. \tag{96}$$

Of course, when $\mathcal{M}$ is compact, the Hodge-de Rham theorem tells us that the set of $L^2$-normalizable harmonic $p$-forms is identical to the $p$th cohomology group of $\mathcal{M}$ with real coefficients:

$$\mathcal{H}_{L^2}^p(\mathcal{M}, g_{\mathcal{M}}) \simeq H^p(\mathcal{M}, \mathbb{R}) \,, \tag{97}$$

regardless of the choice of metric on $\mathcal{M}$. Finiteness of the cohomology groups in (96) then just implies that $\mathcal{M}$ must have finitely many real cohomology classes, which in particular implies that $\mathcal{M}$ must have a finite number of connected components.

However, when $\mathcal{M}$ is non-compact, the dimension of $\mathcal{H}_{L^2}^p(\mathcal{M}, g_{\mathcal{M}})$ can vary greatly depending on the choice of the metric $g_{\mathcal{M}}$.[27] We now demonstrate that the upper-half plane $\mathbb{H}$, as well as the upper-half plane modulo the $T$-transformation $\tau \mapsto \tau + 1$, equipped with the standard hyperbolic metric given in (69), satisfy $\dim \mathcal{H}_{L^2}^1 = \infty$. This gives a divergent entropy contribution, rendering these spaces invalid as consistent quantum gravity moduli spaces.

Since $\mathbb{H}$ can be mapped one-to-one with the hyperbolic unit disk (parametrized by $z$ with $|z| \le 1$) via

$$z = \frac{\tau - i}{\tau + i} \,, \tag{98}$$

the metric (69) in these new coordinates is given as follows:

$$ds^2 = \frac{dz d\bar{z}}{(1 - |z|^2)^2} \,. \tag{99}$$

As mentioned in [127], a 1-form of the form $\omega_n := z^n dz$ for all $n \ge 0$ is harmonic and $L^2$-normalizable. That it is harmonic is clear, since $\omega_n$ is a holomorphic 1-form. Meanwhile, $L^2$-normalizability follows from the fact that $\overline{(*dz)} = (\det(g))^{-1/2} g^{z\bar{z}} d\bar{z} = d\bar{z}$, telling us that all of the $\omega_n$ must be finite-norm states:

$$\int_{|z|\le 1} \omega_n \wedge \overline{(*\omega_n)} = \int_{|z|\le 1} dz d\bar{z} |z|^{2n} = \frac{2\pi}{2n+1} \,. \tag{100}$$

---

[27]For a helpful review of $L^2$-normalizable harmonic forms, see [127].

As for the infinite strip, $\mathbb{H}/\langle T \rangle$, we claim that the 1-forms

$$\omega_n = e^{-\frac{2\pi\tau_2}{n}}\left(\cos(2\pi\tau_1/n)d\tau_1 + \sin(2\pi\tau_1/n)d\tau_2\right), \qquad (101)$$

are all harmonic and $L^2$-normalizable for all $n$. One can explicitly check that these 1-forms are closed and co-closed (and thus harmonic). Moreover, we see that the $\omega_n$ have finite norm due to the cancellation between the factors of $(\det(g))^{-1/2}$ and the inverse metric in the expression for the Hodge dual of a 1-form. Indeed, from $*d\tau_1 = d\tau_2, *d\tau_2 = -d\tau_1$, we see that the norms are $\int_0^\infty d\tau_2 \exp(-4\pi\tau_2/n) = n/4\pi$.

We have seen that we can explicitly rule out our basic non-compactifiable candidate quantum gravity moduli spaces. What can we say in further generality? From a theorem due to J. Lott, it is known that the metric behavior as one approaches the asymptotic boundary $\partial\mathcal{M}$ is completely responsible for whether $\dim\mathcal{H}_{L^2}^p(\mathcal{M}, g_\mathcal{M})$ is finite or not. The precise statement in [128] is as follows:

**Theorem 5.1** (Lott). *Let $\mathcal{M}$ and $\mathcal{N}$ be complete oriented Riemannian manifolds. Suppose that there are compact submanifolds $\mathcal{K} \subset \mathcal{M}$ and $\mathcal{L} \subset \mathcal{N}$ such that $\mathcal{M}\backslash\mathcal{K}$ and $\mathcal{N}\backslash\mathcal{L}$ are isometric. Then $\mathcal{H}_{L^2}^p(\mathcal{M}, g_\mathcal{M})$ and $\mathcal{H}_{L^2}^p(\mathcal{N}, g_\mathcal{N})$ are either both finite dimensional or both infinite dimensional.*

Physically speaking, whether a moduli space $\mathcal{M}$ in a candidate quantum gravity theory leads to an infinite number of 1d ground states therefore depends only on its asymptotic behavior. In general, it has been shown that $\mathcal{H}_{L^2}^p(\mathcal{M}, g_\mathcal{M})$ will be finite, so long as the growth of the volume is not too large. If $\mathcal{M}$ asymptotically approaches a space resembling a cylinder $[0, \infty) \times \partial\mathcal{M}$, then it was proved by Atiyah, Patodi, and Singer that the spaces $\mathcal{H}_{L^2}^p(\mathcal{M}, g_\mathcal{M})$ with compact $\partial\mathcal{M}$ must always be finite-dimensional [129]. This covers the extreme case where the volume growth of $\mathcal{M}$ as small as that of the real line. On the other hand, if the volume growth of $\mathcal{M}$ is large, then there are two results that are useful for our purposes. The first, due to Mazzeo, is as follows [130]:

**Theorem 5.2** (Mazzeo). *Let $\mathcal{M}$ be a $2k$-dimensional complete Riemannian manifold with a metric of the form $g_\mathcal{M} = h/\rho^2$, where $h$ is a smooth metric and $\rho$ is a real-valued function satisfying the following three conditions: $\rho \geq 0$, $\rho^{-1}(0) = \partial\mathcal{M}$, and $d\rho|_{\partial\mathcal{M}} = 0$. Then $\dim\mathcal{H}_{L^2}^k(\mathcal{M}, g_\mathcal{M}) = \infty$.*

Note that the hyperbolic upper half-plane is covered by a special case of Thm. 5.2 where we take $\rho = \tau_2$ and $h = d\tau_1^2 + d\tau_2^2$. These types of hyperbolic metrics enjoy the property that the action of the Hodge star operator on middle-dimensional forms is conformally invariant, making the $L^2$ condition easier to satisfy. Accordingly, as we saw in Sec. 4.3 for $\mathbb{H}$, the hyperbolic spaces covered by Thm. 5.2 have exponential growth in the volume as one approaches the asymptotic boundary.

We have seen that hyperbolic volume growth is generically excluded by Thm. 5.2. However, infinite volume moduli space appears to be allowed. For the example of M-theory on a Klein bottle, where $\mathcal{M} = \mathbb{R} \times \mathbb{R}^+$ with a flat metric, the Euclidean volume growth law $V(D) \sim D^{\dim(\mathcal{M})}$ as a function of the geodesic distance $D$ is realized in quantum gravity. Thus, it is natural to ask the following: Can there be quantum gravity moduli spaces whose volume grows as $\sim D^N$ for $N > \dim(\mathcal{M})$? The answer appears to be negative given the following result due to Dodziuk ([131], see also [132]), which is stated as follows:

**Theorem 5.3** (Dodziuk). *Let $\mathcal{M} = \mathbb{R}^{2k}$ with metric $g = dr^2 + f(r)^2 d\Omega_{S^{2k-1}}$, then $\dim H_{L^2}^k(\mathcal{M}, g_\mathcal{M}) = \infty$ iff $\int_1^\infty dr/f(r) < \infty$.*

Thus, if $f(r) = r^C$, then we have an infinite ground state degeneracy iff $C > 1$. Given Thm. 5.1, this result carries over to any moduli spaces $\mathcal{M}$ which asymptote to $\mathbb{R}_{r>r_*} \times S^{2k-1}$

with the above class of metrics. For moduli spaces with power-law volume growth to be consistent, we find precisely that they must be compactifiable. This rigorously establishes (94) for these classes of infinite-volume moduli spaces. Note that the proof of Thm. 5.3 in [131] only relies on the angular directions parametrizing $S^{2k-1}$, inasmuch as one just requires a smooth space at $r = 0$. In light of Lott's theorem, however, (which incidentally appeared after [131]), and provided that $\mathcal{M}$ takes the form $dr^2 + f(r)^2 \mathrm{Vol}_{\partial \mathcal{M}}$ for some $r > r_*$, Thm. 5.3 carries over just the same, proving that consistent quantum gravity moduli spaces in this class must also be compactifiable.

Note that while Thm. 5.2 and Thm. 5.3 apply to *even*-dimensional candidate moduli spaces, 1d quantum mechanics target spaces are automatically even-dimensional if our vacuum preserves at least four real supercharges. In particular, this is satisfied if our higher-dimensional starting point is any supersymmetric vacuum in $d > 3$ dimensions.[28]

To summarize, we have thus argued from the principle of finiteness that, at least when there is some supersymmetry in the theory under consideration (which is generically expected if we wish to have massless moduli in the first place), the volume growth of the moduli space should be bounded by that of Euclidean space of the same dimension. Otherwise, the partition function of the fully compactified theory would diverge even at zero temperature. This therefore gives a physical, bottom-up motivation (and partial derivation) for our compactifiability conjecture.

## 6 Conclusions

In this paper, we have approached the study of duality groups in effective theories of quantum gravity from several angles. We emphasized the way in which a duality group $\Gamma$, understood to be a discrete, (generically) spontaneously-broken 0-form gauge symmetries, acts on the moduli space of vacua $\mathcal{M}$ of the EFT. Moreover, $\Gamma$ is also seen to act on the entire spectrum of objects in the theory charged under $p$-form gauge symmetry, and these duality actions are intimately connected to the geometry of the moduli space $\mathcal{M}$. Motivated by this, we argued for a relationship between the compactifiability of $\mathcal{M}$, a global geometric property, and the semisimplicity of representations of $\Gamma$, a representation-theoretic property. We finally argued for the compactifiability of $\mathcal{M}$ from the bottom up, relating it to the finiteness of the number of massless states in the fully compactified 1d theory.

From this, a striking picture emerges – it seems that the *existence* of dualities in string theories can potentially be understood from the bottom up. Suppose we are given a consistent EFT coupled to gravity equipped with an exact moduli space parametrized by its massless scalars. We know that (non-gravitational) field theories typically have moduli spaces with non-*negative* curvature; for example, take 4d $\mathcal{N} = 2$ rigid field theories [133]. On the other hand, gravitational theories typically have moduli space space regions with negative curvature. In fact, various Swampland conjectures suggest that $\mathcal{M}$ has non-positive curvature sufficiently far away from positive curvature singularities [2] and that the marked moduli space $\hat{\mathcal{M}}$ has the unique geodesic property [26] (see Sec. 3.4 for more details). In general, it appears that the marked moduli space $\hat{\mathcal{M}}$ has many of the features of manifolds of everywhere non-positive curvature. If the asymptotic curvature of $\hat{\mathcal{M}}$ is in addition *negative*, then it is known that $\hat{\mathcal{M}}$ suffers exponential volume growth (Theorem 13.1 of [134]). To intuitively see this, note that negative curvature pushes nearby geodesics apart, so a simply-connected space with everywhere negative curvature will "spread out" very quickly.

---

[28]For cases with less supersymmetry and with an odd-dimensional moduli space, it should be noted that in certain cases where compactifiability of the candidate moduli space is violated, the spectrum for the Laplacian is gapless for $L^2$-normalizable $(\dim(\mathcal{M}) \pm 1)/2$-forms [130]. It would be interesting to understand if there are physical reasons to exclude such moduli spaces.

We propose in this paper that the moduli space $\mathcal{M}$ should be compactifiable. This principle does not impose significant restrictions on moduli spaces with non-negative curvature, such as those found in field theories, but it provides strong constraints on gravitational theories with moduli spaces of negative curvature. The only way out is that a negatively-curved $\mathcal{M}$ must be a quotient of $\hat{\mathcal{M}}$ by a "large enough" group $\Gamma$ that identifies many fundamental regions under its action in such a way that the exponential volume growth is tamed. This argues for the existence of duality groups for negatively-curved moduli spaces. For *flat* moduli spaces, the volume growth of $\hat{\mathcal{M}}$ is already polynomial, and indeed, such examples need not have dualities (for instance, Type IIA in 10d). Although what we have presented is not a proof, it provides compelling evidence for a general bottom-up argument for duality groups.

This picture is also closely tied to the semisimplicity of duality representations. In particular, for Type IIB in 10d, one needs to quotient the marked moduli space $\hat{\mathcal{M}} = \mathbb{H}$ by a sufficiently large group $\Gamma$ to obtain a compactifiable moduli space $\mathcal{M}$. Such groups $\Gamma$ whose fundamental regions are compactifiable are known in the mathematical literature as *Fuchsian groups*, and they typically include "non-perturbative" duality transformations. It is precisely these transformations that implement the semisimplicity of $\Gamma$. For example, $\mathrm{SL}(2,\mathbb{Z})$ itself is a Fuchsian group, generated by

$$S = \begin{bmatrix} 0 & -1 \\ 1 & 0 \end{bmatrix}, \qquad T = \begin{bmatrix} 1 & 1 \\ 0 & 1 \end{bmatrix}, \tag{102}$$

where $S$ acts via $\tau \mapsto -1/\tau$. Recall that the subgroup generated by $T$ is the upper-triangular subgroup $B$, which is *not* semisimple. It is the inclusion of the "non-perturbative" $S$-transformation that compactifies the moduli space and renders $\mathrm{SL}(2,\mathbb{Z})$ semisimple.

We have painted a relationship between duality and geometry in broad brush strokes, but there are many features that would be interesting to explore further. First of all, we have restricted our attention to the study of duality actions in theories with exact moduli spaces $\mathcal{M}$, which in all known examples require at least 8 supercharges preserved. However, in phenomenologically realistic string compactifications with less supersymmetry, there will typically be a number of scalar fields whose vevs are stabilized by potentials generated by branes, fluxes, and other ingredients. It is of great importance to understand the applicability of the geometric Swampland program in setups with potentials. Progress towards understanding the Distance Conjecture in theories with potentials was made in [135–139], and it would be interesting to extend our analysis of the *global* scalar geometry to these cases as well.

Second, we argued in this paper for the semisimplicity of duality representations from the compactifiability of moduli spaces. However, the semisimplicity property stands as an interesting possible Swampland criterion in its own right. We proposed in Sec. 2.4 that semisimplicity be understood as relating to the fact that a representation be reconstructible from its trace function, which specifies the values of observables in the corresponding physical theory. It would be interesting to further develop this argument, possibly providing an independent Swampland justification for semisimplicity.

Additionally, while we have given strong motivations (and in special cases, proofs) of our compactifiability conjecture using the finiteness of QG amplitudes, it would be nice to rigorously establish the connection between $L^2$-normalizability of complete non-compact moduli spaces and their volume growth more generally.

More broadly, we mostly restricted our attention in this paper to the study of duality actions on charge lattices, which was in turn reduced to the study of complex duality representations. This forgets a vast amount of interesting structure associated to duality actions. For instance, it would be interesting to further study the action of duality groups on torsion abelian charges, as we have done briefly for O3-plane variants in Type IIB. For instance, the semisimplicity criteria proposed could be studied in the context of duality representations over finite fields.

Also interesting is the *nonabelian* duality action on codimension-2 charged objects, including the duality vortices themselves, briefly discussed in Sec. 2.3.2. In several string theory examples, such as the 4d $\mathcal{N} = 2$ hypermultiplet sectors, these adjoint duality "representations" constitute the entire charged spectrum of the theory acted on by duality. To get a taste for the kinds of features one might study, consider the adjoint map ad : $\Gamma \to \text{Aut}(\Gamma)$. We already argued in 3.3 from the marked moduli space conjecture that $Z(\Gamma) = 0$ for hypermultiplet duality groups. However, one could also consider the *cokernel* of the adjoint map, specified by the *outer automorphism* group $\text{Out}(\Gamma)$. It appears that $\text{Out}(\Gamma)$ furnishes additional global 0-form symmetries, which must either get broken or gauged in the physical theory. If these symmetries are gauged then we simply extend the definition of the duality group which, from magnetic completeness, implies that there are more codimension-2 dynamical vortices with monodromy associated with elements $g \in \text{Out}(\Gamma)$. By definition then, if we have identified $\Gamma$ correctly, then $\text{Out}(\Gamma)$ must be broken and it would be interesting to understand the physical mechanisms for how this happens.

Finally, let us consider the study of duality representations from another angle. In quantum gravity, it has long been argued that associated to any gauge symmetry there must be objects transforming in every representation of the gauge group. In principle, one would expect this to hold for duality groups as well. However, we have also seen that the notion of "representation" for a duality group is not always simply a linear representation of it over $\mathbb{R}$ or $\mathbb{C}$ as would be the case for Lie groups. Indeed, since the duality group acts as an outer automorphism group on the various $p$-form gauge groups, and in turn their charge lattices, the notion of "representation" for the duality group ought to be extended to these more general objects. With this in mind, what does it even mean to stipulate completeness of duality representations, and how can we study this in stringy examples? In abstract terms, the duality group acting on the various $p$-form groups (which each may have modified Bianchi identities mixing the various values of $p$) will form an $n$-group structure, and it would be very interesting to develop the precise mathematical notion of completeness for such structures.

## Acknowledgments

We thank Bobby Acharya, Rafael Álvarez-García, Alberto Castellano, Markus Dierigl, Claude LeBrun, Severin Lüst, Miguel Montero, Martin Rocek, Fabian Ruehle, Angel Uranga, Irene Valenzuela, and Timo Weigand for helpful discussions.

**Funding information** The work of DH, SR, CV and KX are supported in part by a grant from the Simons Foundation (602883,CV) and the DellaPietra Foundation. The work of ET is supported in part by the ERC Starting Grant QGuide-101042568 - StG 2021. The authors would like to thank the Simons Summer Workshop 2024 for providing a productive research environment where this project was initiated.

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
