# Peer review of "Finiteness and the Emergence of Dualities"

_SciPost Physics, doi:SciPost Phys. 19, 047 (2025)_

## Round 1 · Referee Report · Anonymous (Referee 1) · 2025-4-30

Strengths

The authors propose a compelling link between finiteness conditions in quantum gravity and the emergence of non-trivial dualities. In particular, they argue that the requirement of finiteness of quantum gravity amplitudes in fully compactified theories (especially with supersymmetry) leads to a bottom-up criterion for the compactifiability of moduli spaces. They further conjecture that this compactifiability condition implies semisimplicity of the duality group's representation on the charge lattice. Their arguments are supported with a diverse array of examples from string theory, including compactifications of type II string theory on Calabi–Yau threefolds.

The paper presents an original idea: that the compactifiability of moduli spaces—a geometric condition—arises naturally from finiteness constraints on QG amplitudes. This connects a seemingly technical requirement (volume growth control) to a rich structure of string theory dualities, providing a new organizing principle within the Swampland program. The authors successfully combine mathematical ideas (orbifold fundamental groups, semisimple representations) with physically motivated conjectures, drawing on tools from algebraic geometry, and representation theory. Despite the abstract nature of the topic, the manuscript is clearly written and includes well-structured examples, particularly the detailed discussion of SL(2,Z) duality in Type IIB string theory. Concepts like the “marked moduli space,” “no-minimum-length conjecture,” and semisimplicity are explained with care and context.

Weaknesses

The “compactifiability” condition is central to the paper but somewhat loosely defined. It would be helpful if the authors offered a more rigorous, possibly geometric or topological, definition—perhaps aligning it more explicitly with concepts from algebraic geometry or metric geometry. Many arguments hinge on supersymmetric settings. The authors mention this caveat, but a discussion of how their framework may (or may not) extend to non-supersymmetric theories would be valuable. Even speculative comments or limitations would help guide future research. The discussion of instantons and duality vortices is interesting but underdeveloped compared to the rest. Since these objects play a key role in moduli space dynamics and corrections to geometry, a deeper treatment would strengthen the overall claims—particularly with respect to how compactifiability relates to instanton corrections.

Report

I recommend the paper for publication in SciPost Physics, pending minor revisions (below). The work is a valuable contribution to ongoing efforts to delineate the landscape and Swampland and suggests new directions to explore the interplay between geometry, duality, and quantum gravity consistency conditions.

Requested changes

• Clarify footnote 1 regarding the role of GL+(2,Z) vs. SL(2,Z).
• Expand the explanation in Section 4.2 on the relationship between harmonic forms and infinite towers in 1D SUSY QM.

Recommendation

Ask for minor revision

---

## Round 1 · Referee Report · Anonymous (Referee 2) · 2025-5-1

Report

The manuscript discusses duality symmetries in superstring theory. It elaborates on their mathematical structure with a focus on their action on the moduli space. It argues for an intriguing relationship between the existence of dualities, a growth criterion on the volume of the moduli space, and the finiteness of quantum gravity amplitudes. The discussion is motivated and supported by multiple supersymmetric string theory examples. The paper is interesting and timely. I recommend publication in SciPost.

Nonetheless, I would suggest the authors to address the following minor comments:

Requested changes

  1. In section 2.1.1 the authors argue for a "complementary definition of the duality group" via the orbifold fundamental group of the scalar manifold. However, they also point out that in general only a quotient $\Gamma'$ of the full duality group $\Gamma$ acts on the moduli space. In this case the orbifold fundamental group of the scalar manifold does not necessarily agree with the full duality group. They also provide explicit string theory examples for this situation later (see section 3.4). I therefore believe that eq. (8) does not provide a sensible definition of the duality group $\Gamma$ unless $\Gamma$ can be uniquely recovered from its quotient $\Gamma'$.

  2. The motivation following the definition in eq. (4) could be made a bit clearer. Are continuous duality groups also conceivable and is focusing on discrete duality groups just a choice of the authors, or are all sensible duality groups necessarily discrete?

  3. There is a small typo below eq. (28): $K_{n(n)}$ should probably read $E_{n(n)}$.

Recommendation

Ask for minor revision

---

## Round 2 · Author Response

We thank the referee for their comments.

---

## Round 2 · List of Changes

We have made the following edits to address the referee's points:
-We fixed the typo in the sentence after eq.(2.1)
-Above eq (2.5), we replaced "...where Γ' is possibly a quotient of the full duality group." with "...where Γ' (which is possibly a quotient of Γ) acts non-trivially on \mathcal{M}." We also replaced "This means that the duality group can in some sense be \textit{identified} with the topology of $\C{M}$" with "This means that the part of the duality group that acts non-trivially on moduli space can in some sense be \textit{identified} with the topology of $\C{M}$"?
-We fixed the typo in the sentence after eq.(2.1)
-Above eq (2.5), we replaced "...where Γ' is possibly a quotient of the full duality group." with "...where Γ' (which is possibly a quotient of Γ) acts non-trivially on \mathcal{M}." We also replaced "This means that the duality group can in some sense be \textit{identified} with the topology of $\C{M}$" with "This means that the part of the duality group that acts non-trivially on moduli space can in some sense be \textit{identified} with the topology of $\C{M}$"?

---

## Editorial Decision

published